# Probabilities of Chat LLMs Are Miscalibrated but Still Predict Correctness on Multiple-Choice Q&A

**Benjamin Plaut**[*]                                                                 *plaut@berkeley.edu*
*Center for Human-Compatible AI*
*UC Berkeley*

**Nguyen X. Khanh**                                                                 *kxnguyen@berkeley.edu*
*Center for Human-Compatible AI*
*UC Berkeley*

**Tu Trinh**                                                                 *tutrinh@berkeley.edu*
*Center for Human-Compatible AI*
*UC Berkeley*

**Reviewed on OpenReview:** <https://openreview.net/forum?id=E6LOh5vz5x>

## Abstract

We evaluate 15 large language models (LLMs) fine-tuned for chat on multiple-choice Q&A. Consistent with prior work, we find that their maximum softmax probabilities (MSPs) are consistently miscalibrated on multiple-choice Q&A. However, those MSPs might still encode useful uncertainty information. Specifically, we hypothesized that wrong answers would be associated with smaller MSPs compared to correct answers. Via rigorous statistical testing, we show that this hypothesis holds for models which perform well on the underlying Q&A task. We also find a strong direct correlation between Q&A accuracy and MSP correctness prediction, while finding no correlation between Q&A accuracy and calibration error. This suggests that within the current fine-tuning paradigm, we can expect correctness prediction but not calibration to improve as LLM capabilities progress. To demonstrate the utility of correctness prediction, we show that when models have the option to abstain, performance can be improved by selectively abstaining based on the MSP of the initial model response, using only a small amount of labeled data to choose the MSP threshold.

## 1 Introduction

Large language models (LLMs) have demonstrated profound capabilities in many domains, but still sometimes generate plausible-sounding false responses (Huang et al., 2023). In one high-profile case, an LLM-based system invented a litany of nonexistent court cases, leading to formal sanctions for two lawyers (Mangan, 2023). Although ongoing work has reduced the rate of these mistakes,[1] LLMs will inevitably face situations that surpass the boundaries of their existing knowledge. In those situations, it is unrealistic to expect these models (or any intelligent agents, including humans) to always make perfect decisions. Rather than confidently misleading users, LLMs should be able to detect unfamiliar situations and act cautiously (e.g., decline to answer).

In this paper, we study whether LLMs can determine the correctness of their own answers to multiple-choice questions. If so, this would directly enable LLMs to decline to answer when they are likely to be incorrect. One natural uncertainty metric for neural networks is the maximum softmax probability (MSP). We study a simple but fundamental question: what does the MSP of an LLM response tell us about whether that

---

[*]Corresponding author
[1]See, for example, <https://huggingface.co/spaces/hallucinations-leaderboard/leaderboard>.

```
Below is a multiple-choice question.  Choose the letter which best answers
the question.  Keep your response as brief as possible; just state the letter
corresponding to your answer, followed by a period, with no explanation.

Question:
In the nitrogen cycle, nitrogen can return to the lithosphere directly from the
atmosphere by
A. lightning.
B. cellular respiration.
C. air pollution.
D. condensation.

Response:
```

Figure 1: A sample question prompt.

response is correct? We are far from the first to study this question, but prior findings are dispersed over a variety of contexts and experimental setups. Our goal is to provide a comprehensive, unified study that both reproduces prior findings and enables novel insights.

## 1.1 Basic setup

We evaluate 15 LLMs fine-tuned for chat (henceforth "chat LLMs" for brevity) on five different Q&A datasets. Figure 1 shows a sample question prompt. The selected LLMs cover a range of sizes, capabilities, and architectures, and include both open-weight and proprietary models. To our knowledge, our work is the most comprehensive study of LLM correctness-awareness.

In multiple-choice Q&A, the MSP is defined as follows. We first compute the probability that the LLM assigns to each answer token (e.g., "A", "B", "C", etc.) and renormalize those probabilities to sum to 1. The MSP is then the maximum of those probabilities. The LLM's response to the question is the token corresponding to the MSP. This approach is consistent with Biderman et al. (2024); Chen et al. (2025); Zhao et al. (2021), among others.

**Calibration.** We first ask whether the MSP is *calibrated* (DeGroot & Fienberg, 1983; Nguyen & O'Connor, 2015), meaning that among responses with an MSP of $p\%$, $p\%$ are correct. Calibrated MSPs enable fully unsupervised abstention policies with theoretical guarantees: a calibrated model that answers only when the MSP is higher than $1 - \varepsilon$ guarantees that the chance of an incorrect answer is at most $\varepsilon$. However, prior work has shown that this approach is not generally viable for chat LLMs (OpenAI, 2023; Zhu et al., 2023): in particular, the MSPs are consistently overconfident.

Our investigation begins by validating those findings on a more diverse set of models and tasks. Furthermore, the comprehensiveness of our study enables robust *cross-model* comparisons that were not possible in prior evaluations that only considered a few models. In particular, we find that calibration does not improve as chat models become more capable.

**Correctness prediction without calibration.** Even if the MSP cannot be directly interpreted as the probability of correctness, it might still be predictive of correctness. As a simplified example, consider a model whose MSP is consistently 0.9 for correct responses and 0.8 for incorrect responses. This model is clearly miscalibrated, but its MSP perfectly predicts correctness.

Through rigorous statistical testing, we demonstrate that the MSPs of chat LLMs can indeed predict correctness.[2] This finding is not surprising given that calibration can often be restored with appropriate rescaling of MSPs (see Section 2 for further discussion). Once again, the more interesting finding is the cross-model comparison. We find that this predictiveness is stronger for models which perform better on the underlying

---

[2]Correctness prediction is measured by the Area Under the Receiver Operating Characteristic curve (AUROC).

Table 1: A summary of the results in our paper. Capability is measured by Q&A accuracy, calibration is measured by expected calibration error, and correctness prediction is measured by AUROC. Some of these findings are more novel than others; see Section 2 for details.

|  | Chat LLMs | Base LLMs |
| --- | --- | --- |
| Calibrated? | ✗ (Figure 2, left) | ✓ (Figure 6, left) |
| Calibration improves with capability? | ✗ (Figure 2, right) | ✓ (Figure 6, right) |
| MSP predicts correctness? | ✓ (Table 2) | ✓ (Table 8) |
| MSP correctness prediction improves with capability? | ✓ (Figure 3) | ✓ (Figure 7) |

Q&A task ($p < 10^{-3}$). In other words, the ability to predict correctness will likely strengthen as the general capabilities of LLMs improve (e.g., by scaling up data and model sizes). The same is not true of calibration, as discussed above. These contrasting results reveal a fundamental dichotomy between two approaches to uncertainty quantification, summarized by Table 1.

**Q&A with abstention.** In addition to demonstrating the predictive power of the MSP and maximum logit, we provide a proof-of-concept for how this information can be leveraged to reduce LLM harm in practice. We analyzed a variant of the original Q&A task where models can also abstain and receive 1 point per correct answer, 0 points per abstention, and $-c$ points per wrong answer. We found that for both $c = 1$ and $c = 2$, selectively abstaining based on the MSP and/or maximum logit led to substantial improvements compared to never abstaining. We used a mere 20 data points per dataset (i.e., 20 randomly selected questions and their answers) to select the abstention threshold.

**Base models.** Although our focus is chat LLMs, we run the same experiments for base (i.e., non-fine-tuned) models. Consistent with prior work (Kadavath et al., 2022), we find that the base models are much better calibrated than the chat models, and the calibration of base models *does* improve as Q&A accuracy improves. These positive calibration results suggest that the MSP will also predict correctness, which we confirm rigorously.

**In summary, our key results are the following:**

1. The MSPs of chat LLMs are miscalibrated, and this does *not* improve as model capabilities improve.
2. The MSPs of chat LLMs still predict correctness, and this *does* improve as model capabilities improve.
3. A small amount of labeled data can translate correctness prediction into an effective abstention method.

The paper proceeds as follows. Section 2 discusses related work and Section 3 covers our general experimental setup. Sections 4, 5, and 6 present our results for chat LLMs on calibration, correctness prediction, and Q&A-with-abstention, respectively. Section 7 covers the analogous results for base models. Appendix A provides more detail on the above results. The remaining appendices cover 5-shot prompting experiments (which produce similar results), other measures of uncertainty (margin and entropy), a post-hoc calibration experiment, and dataset-level analysis.

## 2 Related work

**Key comparison: Kadavath et al. (2022).** The most relevant prior paper is Kadavath et al. (2022), who studied LLM correctness-awareness in a variety of domains. There are three key differences between their work and ours.

The first is that *they primarily study base LLMs*. In particular, their well-known findings that (1) LLMs are well-calibrated and (2) calibration improves further as model size/capability increases apply only to base models. In this way, our work complements theirs by showing that their finding of good calibration fails to

generalize to LLMs fine-tuned for chat. Correctness-awareness may also be more important for fine-tuned models, since the casual user is less likely to interact with base LLMs.

The second is that *they only studied MSP calibration and not MSP correctness prediction.*[3] This may be because good calibration directly yields theoretically grounded correctness prediction, so for base models, good calibration may suffice. However, our finding that chat LLMs have miscalibrated MSPs motivates the separate question of whether MSPs can predict correctness.

The third is *comprehensiveness.* They only tested a single series of models, while we test 6 series of models (or 8, depending on how one counts) and 15 models total. Our comprehensiveness crucially enables cross-model comparisons, as discussed in Section 1. In particular, we have statistical evidence that the correlation between correctness prediction and Q&A accuracy (and the lack of a correlation between calibration and Q&A accuracy) may extend to models that do not even exist yet. In contrast, it is harder to claim that the findings of Kadavath et al. (2022) generalize to other models, since they essentially have a sample size of one.

Overall, our work complements theirs. Viewing our work and theirs side-by-side suggests that fine-tuning degrades calibration of LLMs and this effect is not mitigated as models become more capable. However, this procedure only *distorts* rather than erases uncertainty information in LLMs, and that uncertainty information *does* become more useful as models become more capable.

**LLM calibration.** Uncertainty quantification in LLMs is a very active area and a full survey is beyond the scope of this paper; we direct the interested reader to Geng et al. (2024). Several prior papers have found evidence that chat LLMs are miscalibrated (He et al., 2023; OpenAI, 2023; Zhu et al., 2023). These papers also showed that the calibration curves of chat LLMs are roughly monotone (e.g., Figure 2 left), which suggests – but does not prove – that the MSP may predict correctness even though it is miscalibrated.

However, these papers generally only study one model (or at most a few models), and each paper uses different experimental conditions. In contrast, we present a comprehensive and unified evaluation of the MSP calibration and correctness-awareness of 15 chat LLMs. We also mention the simultaneous work by Xiao et al. (2025), which studies the calibration of four chat LLMs and finds the same overconfidence.

**Post-hoc calibration.** The miscalibration of chat LLMs has motivated the development of methods to restore calibration by rescaling the MSP in some way. Temperature scaling and variants thereof are especially popular, which have been shown to improve calibration without harming performance (Shen et al., 2024; Xie et al., 2024). There also exist generation-based methods. For example, Zhang et al. (2024) replace the model's chosen response with "All other options are wrong" and test whether the model still selects that option. Ulmer et al. (2024) trained a predictor based on input-output pairs and calibration targets. Xiao et al. (2025) proposed a calibration-aware fine-tuning method applied after standard fine-tuning which restores calibration.

The success of post-hoc calibration methods aligns with our finding that MSPs can predict correctness even when poorly calibrated. In both cases, the key insight is that although the fine-tuning process disrupts calibration, the MSPs of chat LLMs retain an underlying uncertainty signal which can be recovered.

**Verbalized uncertainty in LLMs.** An alternative way to obtain uncertainty estimates from LLMs is to prompt them directly. One benefit of this approach is that it requires no access to the internals of the model. However, this approach has produced mixed results: LLMs can sometimes verbalize calibrated confidence levels (Lin et al., 2022a; Tian et al., 2023), but can also be highly overconfident (Xiong et al., 2024). Interestingly, Xiong et al. (2024) found that LLMs typically state confidence values in the range of 80-100%, usually in multiples of 5, potentially in imitation of how humans discuss confidence levels. Nevertheless, prompting strategies remain an important tool for uncertainty quantification, along with measures based on the internal state (such as MSP).

**Training LLMs to abstain.** Another line of work has fine-tuned LLMs to predict the correctness of their answers (Kadavath et al., 2022; Yin et al., 2023; Zhang et al., 2023). This approach has a different focus than our work: our goal is to understand the fundamental relationship between MSPs and correctness, not to design a state-of-the-art abstention method. Our Q&A-with-abstention experiments are intended as a

---

[3]They did study correctness prediction in the different context of training LLMs to abstain. We discuss those results separately below.

simple proof of concept of our correctness prediction findings. In other words, our primary contribution is scientific, not methodological.

**Abstaining based on the MSP.** The idea of abstaining based on the MSP was originally introduced by Chow (1970) in the context of pattern recognition. This technique was recently explored for LLMs by Gupta et al. (2024), although their setting is different. Also, their experiments only use FLAN-T5 models. In contrast, we test 15 different LLMs, which enables the cross-model comparisons previously discussed.

**Beyond LLMs.** The MSP has been used for anomaly/out-of-distribution detection in a variety of other contexts, including coreference resolution tasks (Nguyen & O'Connor, 2015), pre-trained BERT models (Hendrycks et al., 2020), and image classification (Hendrycks et al., 2022; Hendrycks & Gimpel, 2017).

## 3 Experimental setup

This section presents our general experimental setup. Elements of the setup that are specific to calibration, correctness prediction, or abstention are discussed in Sections 4, 5, and 6, respectively. Our code can be found at https://github.com/bplaut/llm-calibration-and-correctness-prediction.

**Multiple-choice Q&A.** We chose to study multiple-choice Q&A because there is exactly one correct answer. This allows us to study the core hypothesis of whether MSPs are predictive of correctness without potential confounders such as degrees of correctness or multiple valid phrasings of the same correct answer.

**Datasets.** We based our experimental framework on the original Hugging Face LLM leaderboard (Beeching et al., 2023). We used all five multiple-choice Q&A datasets[4] from that leaderboard: ARC-Challenge (Clark et al., 2018), HellaSwag (Zellers et al., 2019), MMLU (Hendrycks et al., 2021), TruthfulQA (Lin et al., 2022b), and WinoGrande (Sakaguchi et al., 2021). We randomly sampled 6,000 questions from HellaSwag, MMLU, and WinoGrande. ARC-Challenge and TruthfulQA only have 2,590 and 817 questions, respectively, so we used all of those questions. The 6,000 number was chosen to make the experiment duration manageable.

**Prompting style.** To test our hypothesis in the simplest possible setting, we used a plain zero-shot prompting style. For comparison, we also ran the experiments with 5-shot prompting and obtained similar results (Appendix B). We also used two different phrasings to ensure that our results were not an artifact of the specific phrasing. One phrasing is shown in Figure 1 and the other appears in Appendix A.

**Models.** We tested 15 LLMs: 13 open-weight and two proprietary. The open-weight LLMs were chosen based on a combination of performance on the aforementioned leaderboard and the number of downloads on Hugging Face. The open-weight models we selected are Falcon (7B and 40B) (Almazrouei et al., 2023), Llama 2 (7B, 70B) (Touvron et al., 2023), Llama 3.0 (8B, 70B) and Llama 3.1 (8B, 70B) (AI@Meta, 2024), Mistral 7B v0.2 (Jiang et al., 2023), Mixtral 8x7B (Jiang et al., 2024), SOLAR 10.7B (Kim et al., 2023), and Yi (6B and 34B) (01-ai, 2023). All of the open-weight LLMs were accessed through the Hugging Face interface and were run with dynamic 4-bit quantization, which has been shown to preserve performance while reducing computational requirements (Dettmers et al., 2023). We tested both fine-tuned "chat" and non-fine-tuned "base" versions of these 13 open-weight models. Unless we specify otherwise, the reader should assume we are referring to the "chat" versions of these models. The experiments on open-weight LLMs took about 2000 GPU-hours using NVIDIA RTX A6000 GPUs.

We also tested two proprietary LLMs for which we could obtain softmax probabilities through an API: OpenAI's GPT-3.5 Turbo (Brown et al., 2020; Ouyang et al., 2022) and GPT-4o (OpenAI, 2024). The OpenAI API does not provide pre-softmax logits, so we could not compute Max Logit AUROCs for those two models.[5] We also did not have access to non-fine-tuned counterparts of these models. These experiments took about a day and cost about $110.

**Aggregating results across datasets.** Grouping the questions from all datasets together to compute a single AUROC per model would undervalue datasets with fewer questions. Instead, we computed a separate AUROC for each available combination of model, dataset, prompt phrasing, and classifier (MSP vs Max

---

[4]The leaderboard also includes the GSM8k dataset, which we excluded since it is not multiple-choice.

[5]We also could not test "reasoning" models such as o1 and o3, since the API provides neither logits nor probabilities for those models.

Logit).[6] All in all, we recorded 280 AUROC data points (15 models $\times$ 5 datasets $\times$ 2 phrasings $\times$ 2 classifiers, excluding Max Logit for OpenAI models) and 150 Q&A accuracy data points (15 models $\times$ 5 datasets $\times$ 2 phrasings) over a total of 642,210 prompts (15 models $\times$ 21,407 questions across datasets $\times$ 2 phrasings). We then calculated per-model unweighted averages to get the results in Table 2.

**Computing MSP and Max Logit.** Let $\mathcal{V}$ be a set of tokens (a vocabulary) and let $\boldsymbol{x}$ be a sequence of tokens from $\mathcal{V}$. For each token $y \in \mathcal{V}$ and prefix $\boldsymbol{x}$, an LLM computes a logit $L(y \mid \boldsymbol{x})$. A softmax function is applied to the logits to derive the probability of $y$ being the next token:

$$P(y \mid \boldsymbol{x}) = \frac{\exp(L(y \mid \boldsymbol{x}))}{\sum_{z \in \mathcal{V}} \exp(L(z \mid \boldsymbol{x}))}$$

In our experiments, we formulated each question as a prompt $\boldsymbol{x}$ as in Figure 1. Let $\mathcal{T}$ be the set of possible answer tokens, i.e., $\mathcal{T} = \{A, B, \cdots\}$. We then computed $P(y \mid \boldsymbol{x})$ and $L(y \mid \boldsymbol{x})$ for each $y \in \mathcal{T}$, i.e., the probability and logit for $y$ being the first token of the response.[7] The LLM's answer is the token with the maximum such probability and logit, i.e., $\arg\max_{y \in \mathcal{T}} P(y \mid \boldsymbol{x}) = \arg\max_{y \in \mathcal{T}} L(y \mid \boldsymbol{x})$. The MSP and Max Logit are the probability and logit associated with that token, respectively. The probabilities were also renormalized to sum to 1 over tokens in $\mathcal{T}$. Formally,

$$\mathrm{MSP}(\boldsymbol{x}) = \frac{\max_{y \in \mathcal{T}} P(y \mid \boldsymbol{x})}{\sum_{y \in \mathcal{T}} P(y \mid \boldsymbol{x})} \qquad\qquad \mathrm{Max\ Logit}(\boldsymbol{x}) = \max_{y \in \mathcal{T}} L(y \mid \boldsymbol{x})$$

## 4 Calibration

We first asked whether the MSPs of chat LLMs are calibrated. For each possible combination of LLM, dataset, and prompt phrasing, we performed two analyses. First, we computed each model's calibration curve as follows. For each model, we divided the range of possible MSPs into 10 quantile bins, i.e., each containing the same number of data points. Then for each bin, we computed the average MSP and the fraction of correct responses in that bin. Figure 2 (left) displays each model's calibration curve. Most models exhibit clear overconfidence: the MSP is consistently much larger than the fraction of correct responses. For example, most models produce MSPs above 0.95 even when they are correct only 60% of the time.

This shows that most models are miscalibrated, but how does the level of miscalibration vary between models? To answer this question, we computed each model's total calibration error, equal to the mean over bins of the absolute difference between the average MSP and the fraction of correct answers. To avoid downweighting smaller datasets, we first computed the calibration error for each model-dataset pair and then computed per-model unweighted averages across the five datasets (and two prompt phrasings). We then plot each model's calibration error vs its overall Q&A accuracy. This allows us to see whether calibration error goes down (or up) as models become more powerful.

Figure 2 (right) shows that the answer is no. Although a slight downward trend exists, it is statistically insignificant. More formally, the coefficient of determination was $R^2 = 0.07$ ($p = 0.33$), meaning that Q&A accuracy can explain only 7% of the variance in calibration error.[8] As such, we should not expect chat LLMs to become more calibrated as their capabilities grow. Despite this, we will see in Section 5 that we *can* expect MSPs to become increasingly effective at predicting correctness, even as their calibration does not improve.

## 5 Correctness prediction without calibration

Before presenting said results, we must define correctness prediction. Correctness prediction is a binary classification task: given a multiple-choice question and the LLM's response, predict whether the response is correct. We study two classifiers for this task: the MSP classifier (predict correctness iff the MSP exceeds

---

[6] AUROC was computed by the Python module sklearn.

[7] A few models always begin responses with a dummy token, so for those models, we used the second token of the response.

[8] Coefficients of determination and $p$ values for cross-model correlations (e.g., Figures 2 right and 3) were computed via standard linear regression.

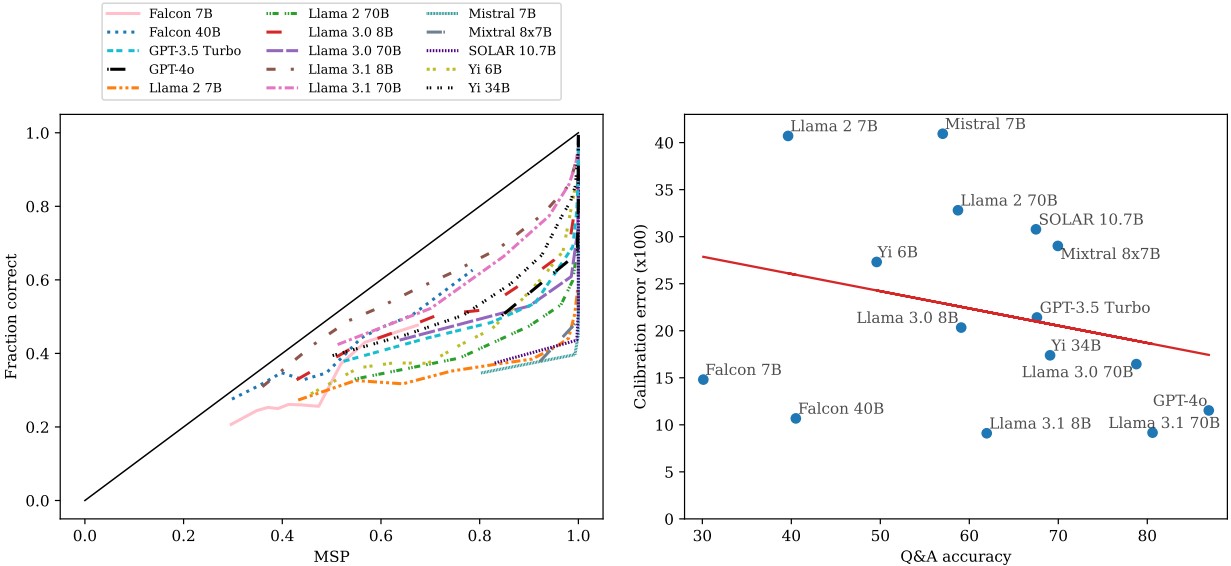

Figure 2: **Left:** The calibration curve for each model. Most models exhibit clear overconfidence: the MSP is larger than the true fraction of correct responses. **Right:** The average calibration error per model (scaled by 100 to improve readability) vs Q&A accuracy. There is no statistical evidence for a correlation between calibration error and Q&A accuracy ($R^2 = 0.07, p = 0.33$). The precise calibration error values can be found in the appendix (Table 6).

some threshold) and the Max Logit[9] classifier (predict correctness iff the maximum pre-softmax logit exceeds some threshold). We hypothesized that the MSP and Max Logit classifiers would (statistically) outperform random chance on this classification task.

Note that we are not training a new binary classifier for this task: we study the performance of the MSP and Max Logit "out of the box", since our goal is to understand the innate properties of LLMs.

## 5.1 Methodology

**AUROC.** Performance on a binary classification task is often measured by the Area Under the Receiver Operating Characteristic curve (AUROC) (Bradley, 1997). The AUROC of a binary classifier ranges from 0% to 100%, where 0% corresponds to getting every prediction wrong, 50% is random chance, and 100% is perfect classification. AUROC is also equivalent to the probability that a randomly chosen positive instance is ranked higher than a randomly chosen negative instance. Conveniently, AUROC is threshold-independent in that it captures the model's performance across the entire range of possible thresholds.[10]

We computed the AUROC for each available combination of LLM, dataset, prompt phrasing, and classifier (MSP or Max Logit). We could not compute Max Logit AUROCs for the OpenAI models (GPT-3.5 Turbo and GPT-4o) because the OpenAI API only provides softmax probabilities and not pre-softmax logits. This resulted in 280 AUROC data points: 150 for MSP and 130 for Max Logit. For each model-classifier pair, we computed an unweighted average across 10 data points (five datasets × two prompt phrasings) to obtain the values in Table 2.

**Statistical significance.** To determine whether our AUROC results were statistically significant, we used the Mann-Whitney $U$ test (MWU) (Mann & Whitney, 1947; Wilcoxon, 1945). The MWU directly tests the null hypothesis that a classifier's true AUROC is 50% (i.e., random guessing). For us, a significant MWU affirms the hypothesis that our classifier can distinguish between (1) questions where the LLM answered correctly and (2) questions where the LLM answered incorrectly.

---

[9]Max Logit has no notion of calibration, since it is not a probability.
[10]Note that calibration error is also threshold independent.

For each available combination of model, dataset, prompt phrasing, and classifier, we tested the null hypothesis that the true AUROC was equal to 50%. This resulted in 280 MWUs. Table 2 reports the number of $p$-values which were below $10^{-4}$ for each model-classifier pair. The threshold of $\alpha = 10^{-4}$ accounts for a Bonferroni correction (Bonferroni, 1936), which is applied when performing multiple hypothesis tests (in our case, 280) to ensure that the chance of falsely rejecting *any* null hypothesis is small. Starting from the standard threshold of $\alpha = 0.05$, the Bonferroni correction yields $\alpha = 0.05/280 \approx 1.8 \times 10^{-4}$. We use the stricter threshold of $10^{-4}$ for simplicity.[11]

## 5.2 Results

**Key results.** Among these 280 data points, AUROC outperformed random chance with $p < 10^{-4}$ in 245 cases. When the Falcon and Llama 2 models are excluded, that statistic improves to 196/200. These results demonstrate that the MSP and Max Logit are statistically predictive of correctness (except for possibly the weakest models).

Our more exciting finding is a strong direct correlation between average Q&A accuracy and average AUROC: the coefficients of determination for MSP AUROC and Max Logit AUROC were $R^2 = 0.94$ and $R^2 = 0.71$, both with $p < 10^{-3}$ (Figure 3). This finding suggests that as chat LLMs become more powerful, the uncertainty information present in MSPs actually becomes more refined. In fact, that might be an understatement given that Q&A accuracy accounts for a remarkable 94% of the variance in MSP AUROC. In other words, a model's Q&A accuracy almost entirely determines how well that model's MSP can predict correctness, regardless of any other factors like architecture, size, etc. In contrast, we found no evidence that calibration error will similarly improve (Figure 2, right).

One theory for the correlation between model capability and correctness prediction is the following:

1. For base models, calibration does improve with capability (Kadavath et al., 2022 and Section 7).
2. Post-hoc calibration methods like Platt scaling (Platt, 2000) can improve calibration for miscalibrated LLMs (Shen et al., 2024; Ulmer et al., 2024; Xiao et al., 2025; Xie et al., 2024; Zhang et al., 2024).
3. Since AUROC depends only on the ordering of scores (in our case, the MSP or maximum logit) and Platt scaling preserves order, Points 1 and 2 suggest that correctness prediction AUROC will also improve with capability.

However, this line of reasoning does not explain the impressive strength of the correlation ($R^2 = 0.94$). Indeed, the correlation we find between Q&A accuracy and calibration error in base models in Figure 7 is much weaker than this ($R^2 = 0.51, p = 0.006$). We find this intriguing.

**An outlier dataset: WinoGrande.** WinoGrande was by far the hardest dataset for our correctness prediction task (Table 3). Our best hypothesis for this discrepancy is that WinoGrande is intentionally adversarial and tries to "trick" the model. An illustrative question from this dataset is "Neil told Craig that he has to take care of the child for the day because __ did it last time." Even for some humans, it could be unclear whether Neil is assuming responsibility or assigning responsibility. One wrinkle is that the Q&A accuracy on WinoGrande is comparable to other datasets, so it is not the case that this dataset is "harder" in general: it is harder only for predicting correctness. Despite the average MSP AUROC of 60.8% for WinoGrande, Llama 3.1 70B and GPT-4o still achieved AUROCs of 75.8% and 78.3% respectively on this dataset (Table 25), suggesting that this difficulty is surmountable for capable models.

**Minimal correlation between model size and AUROC.** The coefficients of determination between model size and AUROC were $R^2 = 0.35$ ($p = 0.03$) and $R^2 = 0.16$ ($p = 0.17$) for MSP and Max Logit, respectively (Figure 4). It is unsurprising that some correlation exists, due to the known correlation between size and model capabilities (e.g., Q&A accuracy) and our correlation between Q&A accuracy and AUROC. However, the relatively weak correlation between model size and AUROC suggests that it is actually Q&A accuracy and not model size that matters.

---

[11] We handle the cross-model comparisons separately since we test only 18 cross-model hypotheses (one in Figure 2, two in Figure 3, two in Figure 4, and 13 more in the appendix). As such, we use $\alpha = 0.05/18 \approx 0.003$ for those comparisons. For simplicity, we use $\alpha = 10^{-3}$ for cross-model comparisons.

Table 2: Main AUROC results. AUROC and Q&A values are percentages, averaged over ten data points (five datasets and two phrasings). The $p < 10^{-4}$ columns indicate how many of those ten data points yielded $p$-values below $10^{-4}$ for the null hypothesis that AUROC $= 50\%$. The $p$-values are from the Mann-Whitney $U$ test; see Section 5.1 for details.

| LLM | Q&A accuracy | MSP | | Max Logit | |
|---|---|---|---|---|---|
| | | AUROC | $p < 10^{-4}$ | AUROC | $p < 10^{-4}$ |
| Falcon 7B | 30.1 | 52.0 | 1/10 | 51.9 | 1/10 |
| Falcon 40B | 40.5 | 60.2 | 7/10 | 58.6 | 7/10 |
| Llama 2 7B | 39.7 | 58.6 | 6/10 | 57.8 | 7/10 |
| Llama 2 70B | 58.7 | 71.1 | 10/10 | 66.9 | 9/10 |
| Llama 3.0 8B | 59.1 | 71.2 | 10/10 | 68.9 | 10/10 |
| Llama 3.0 70B | 78.8 | 81.3 | 10/10 | 68.4 | 9/10 |
| Llama 3.1 8B | 62.0 | 72.9 | 10/10 | 67.3 | 10/10 |
| Llama 3.1 70B | 80.6 | 83.4 | 10/10 | 69.6 | 10/10 |
| Mistral 7B | 57.0 | 66.1 | 10/10 | 62.8 | 10/10 |
| Mixtral 8x7B | 69.9 | 70.0 | 10/10 | 60.5 | 8/10 |
| SOLAR 10.7B | 67.5 | 71.8 | 10/10 | 66.8 | 10/10 |
| Yi 6B | 49.6 | 67.0 | 10/10 | 60.4 | 10/10 |
| Yi 34B | 69.1 | 75.8 | 10/10 | 69.0 | 10/10 |
| GPT-3.5 Turbo | 67.5 | 76.1 | 10/10 | − | − |
| GPT-4o | 86.9 | 85.3 | 10/10 | − | − |

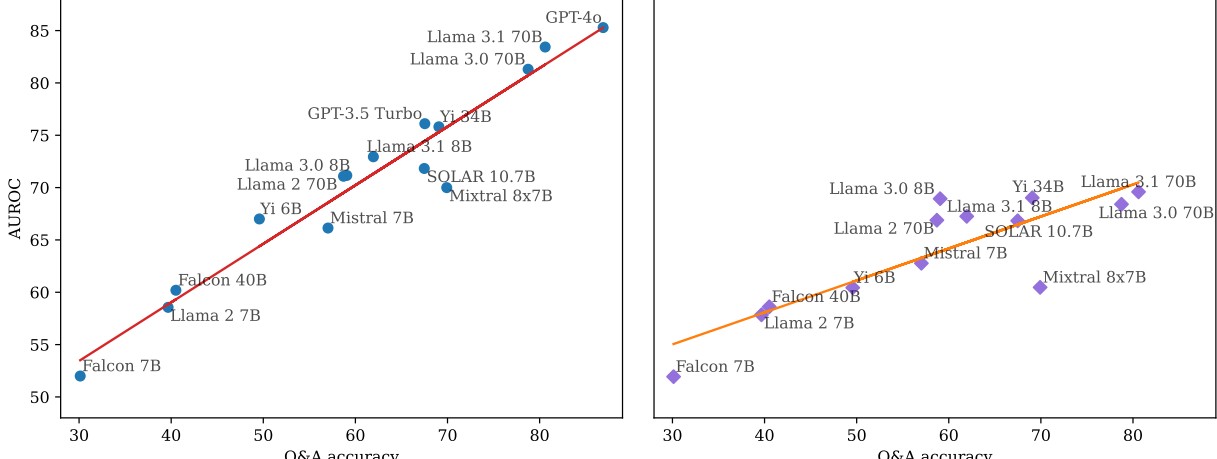

Figure 3: Average AUROC vs average Q&A accuracy for the MSP (left) and Max Logit (right). These plots use the same data as Table 2. The coefficients of determination for MSP and Max Logit were $R^2 = 0.94$ and $R^2 = 0.71$ respectively, both with $p < 10^{-3}$, indicating strong correlations.

**Maximum vs margin.** In the main body of the paper, we only consider the maximum probability and maximum logit. However, there are other ways to measure uncertainty as well. In Appendix C, we also consider the margin (difference between the top two probabilities/logits) and entropy (only for probabilities). One interesting finding is that for logits specifically, both the AUROC values and the strength of the correlation with Q&A accuracy are much higher for the margin compared to the maximum. This effect does not exist for probabilities. Our theory is that the probabilities are already normalized, while the logits are not, and the margin sort of functions as normalization.

Table 3: Average Q&A accuracy, AUROCs, and calibration error per dataset. All values are averaged over the 15 models and two prompts.

|  | Q&A accuracy | MSP AUROC | Max Logit AUROC | Calibration Error (x100) |
|---|---|---|---|---|
| ARC-Challenge | 71.8 | 78.1 | 70.0 | 13.6 |
| HellaSwag | 60.1 | 71.0 | 63.9 | 19.6 |
| MMLU | 56.9 | 73.3 | 66.8 | 24.1 |
| TruthfulQA | 51.4 | 71.0 | 63.3 | 30.6 |
| WinoGrande | 60.8 | 60.8 | 54.8 | 24.2 |

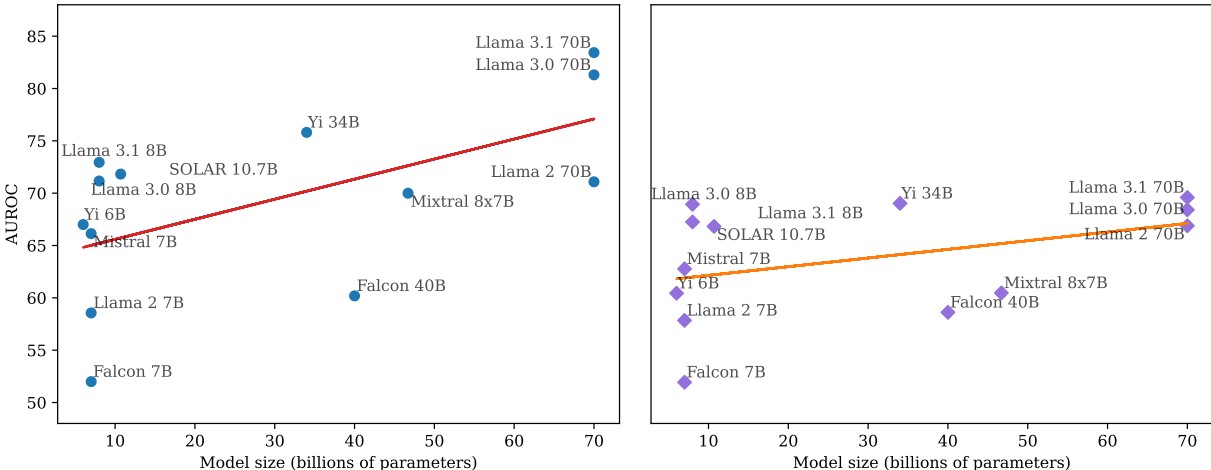

Figure 4: AUROC vs model size for MSP (left) and Max Logit (right). The coefficients of determination for MSP and Max Logit were $R^2 = 0.35$ ($p = 0.03$) and $R^2 = 0.16$ ($p = 0.17$) respectively. GPT-3.5 Turbo and GPT-4o were excluded since their sizes are unknown.

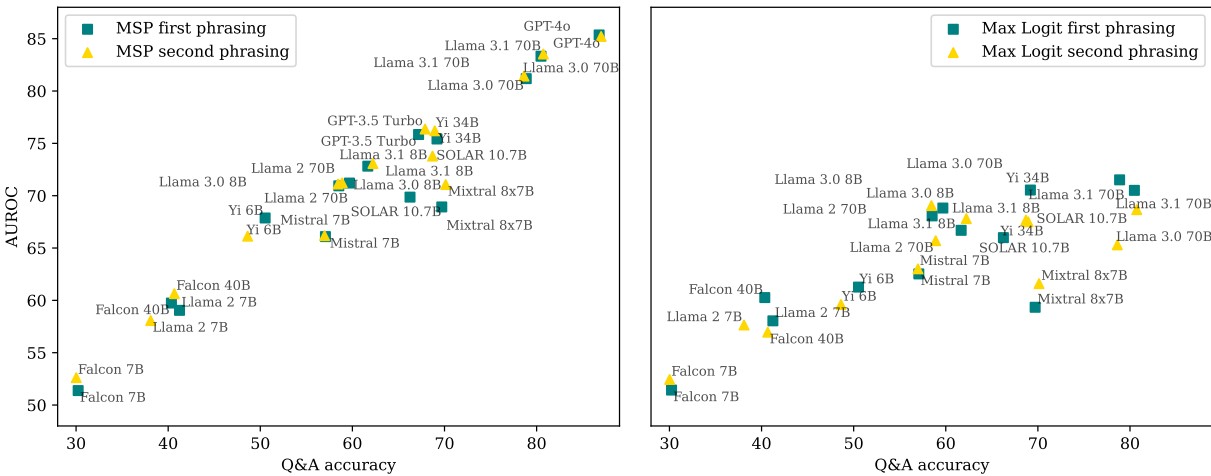

Figure 5: Average AUROC vs Q&A accuracy based on prompt phrasing (the two phrasings can be found in Figures 1 and 8). All values are averaged over the five datasets.

**Prompt phrasing had minimal impact.** The two prompt phrasings (Figures 1 and 8) yielded similar results (Figure 5). This suggests that our results are robust to minor modifications to the prompt.

# 6 Proof-of-concept: reducing wrong answers by abstention

In Section 5, we showed that the MSP and maximum logit can predict correctness. To illustrate the utility of this finding, we now revisit the original Q&A task but allow models to selectively abstain based on the MSP or Max Logit.

## 6.1 Methodology

These experiments use the same data as the AUROC experiments, but the data is analyzed differently. For each classifier (MSP or Max Logit) and a given threshold, we conducted the following analysis. First, we computed the classifier value (MSP or maximum logit) based on the initial LLM response, the same way we did in our AUROC experiments. If the classifier value was below the threshold, we recorded the model's answer as "abstain" and otherwise recorded the original answer. We awarded 1 point per correct answer, 0 points per abstention, and $-c$ points per wrong answer, normalized by the total number of questions. We performed this analysis for $c = 1$ ("balanced score") and $c = 2$ ("conservative score"). For $c = 1$, the benefit of a correct answer equals the cost of a wrong answer. However, wrong answers are often much worse (e.g., medical diagnoses), justifying $c = 2$. Our experiment design was partly inspired by Kang et al. (2024), who used an even more extreme penalty of $c = 4$.

**Choosing the threshold.** Unlike our AUROC results, here we must choose a specific threshold for whether the model should abstain. To do so, we randomly selected a training set of $k$ questions per dataset. Then for each model, we chose the threshold which performed best on those $5k$ questions. We use a single threshold for each model across all datasets in order to make our method more robust. We discovered that $k = 20$ performed almost as well as using half of all questions. Figure 10 in the appendix visualizes the Q&A-with-abstention score as a function of $k$. Unless otherwise specified, all figures and tables use $k = 20$.

## 6.2 Results

Table 4: Results on Q&A with abstention. "Balanced" and "conservative" correspond to -1 and -2 points per wrong answer, respectively. Correct answers and abstentions are always worth +1 and 0 points, respectively. The total number of points is divided by the total number of questions (then scaled up by 100 for readability) to obtain the values shown in the table. We highlight the best method for each model.

| | Balanced | | | Conservative | | |
| LLM | No abstain | MSP | Max Logit | No abstain | MSP | Max Logit |
| --- | --- | --- | --- | --- | --- | --- |
| Falcon 7B | −39.8 | **−0.7** | −2.2 | −109.7 | **−7.8** | −8.0 |
| Falcon 40B | −19.0 | **2.0** | 0.9 | −78.4 | 0.0 | **0.1** |
| Llama 2 7B | −20.7 | 0.1 | **0.6** | −81.1 | **−1.3** | −5.0 |
| Llama 2 70B | 17.4 | 19.9 | **20.7** | −23.9 | **6.5** | 3.5 |
| Llama 3.0 8B | 18.1 | **23.9** | 22.8 | −22.9 | **11.9** | 5.7 |
| Llama 3.0 70B | 57.5 | **58.7** | 57.5 | 36.3 | **46.3** | 41.4 |
| Llama 3.1 8B | 23.9 | **29.2** | 20.5 | −14.2 | **17.6** | 10.2 |
| Llama 3.1 70B | 61.2 | **62.8** | 61.2 | 41.8 | **49.9** | 42.8 |
| Mistral 7B | 14.0 | 15.5 | **16.8** | −29.0 | 0.0 | **1.9** |
| Mixtral 8x7B | 39.8 | **40.3** | 39.8 | 9.8 | **15.8** | 8.2 |
| SOLAR 10.7B | 34.9 | **36.7** | 34.9 | 2.4 | **12.4** | 8.7 |
| Yi 6B | −0.9 | **9.6** | 5.0 | −51.4 | **5.8** | −0.2 |
| Yi 34B | 38.1 | **41.2** | 38.2 | 7.1 | **22.5** | 20.4 |
| GPT-3.5 Turbo | 35.1 | **36.3** | – | 2.6 | **28.3** | – |
| GPT-4o | 73.8 | **73.9** | – | 60.7 | **61.1** | – |

For each combination of LLM, classifier, and $c \in \{1, 2\}$, Table 4 reports the scores obtained by the base LLM and by our method on the test set, where our method used the threshold determined by the training set. Figure 9 shows each model's score across the entire range of possible thresholds. Our method outperformed or matched the base LLM in all conditions and substantially outperformed the base LLM on the conservative score metric.

As expected, models with low initial scores exhibited the most dramatic improvements. For example, any model with a negative initial score can trivially improve to 0 by abstaining on every question. More generally, the higher the fraction of correct answers, the more likely the model is to accidentally abstain on a correct answer. As a result, it is unsurprising that models with high initial scores showed more modest improvements and lower abstention rates.

**Abstention frequency.** In line with the reasoning above, some models do abstain quite frequently, with some of the weakest models reaching nearly 100% abstention rates (Table 7 in the appendix). One may be concerned that excessively frequent abstention could render a model unusable. However, we argue that excessively frequent wrong answers would render a system not only unusable but actively harmful. If a model is likely to be wrong more often than not, perhaps it is appropriate for the model to always abstain.

Overall, our Q&A-with-abstention results show how the uncertainty signals from softmax probabilities and/or logits can be leveraged to improve performance on practical language tasks.

## 7 Experiments on base LLMs

Our paper focuses on chat LLMs since the questions we study are relatively better understood for base LLMs (due to Kadavath et al., 2022 in particular). However, given our goal of providing a comprehensive study of correctness-awareness, we ran the same suite of experiments and same analysis (calibration, AUROC, and Q&A-with-abstention) for the base LLMs. This section overviews our results for the base models, with details appearing in Appendix A.3. The OpenAI models are excluded from these experiments since we do not have access to their corresponding base models.

The base models were significantly worse at responding in the correct format. That is, it was not uncommon for a model to assign very small probabilities to *all* of the possible answer tokens (Table 5). The answer token probabilities were normalized to 1 before analysis (see Section 3), but we think the base models' results should be regarded with mild skepticism. That said, our findings do align with Kadavath et al. (2022).

Table 5: Median unnormalized MSPs for base LLMs.

| LLM | Median unnormalized MSP |
|---|---|
| Falcon 7B | 0.002 |
| Falcon 40B | 0.023 |
| Llama 2 7B | 0.0037 |
| Llama 2 70B | 0.020 |
| Llama 3.0 8B | 0.099 |
| Llama 3.0 70B | 0.102 |
| Llama 3.1 8B | 0.124 |
| Llama 3.1 70B | 0.078 |
| Mistral 7B | 0.285 |
| Mixtral 8x7B | 0.187 |
| SOLAR 10.7B | 0.020 |
| Yi 6B | 0.182 |
| Yi 34B | 0.005 |

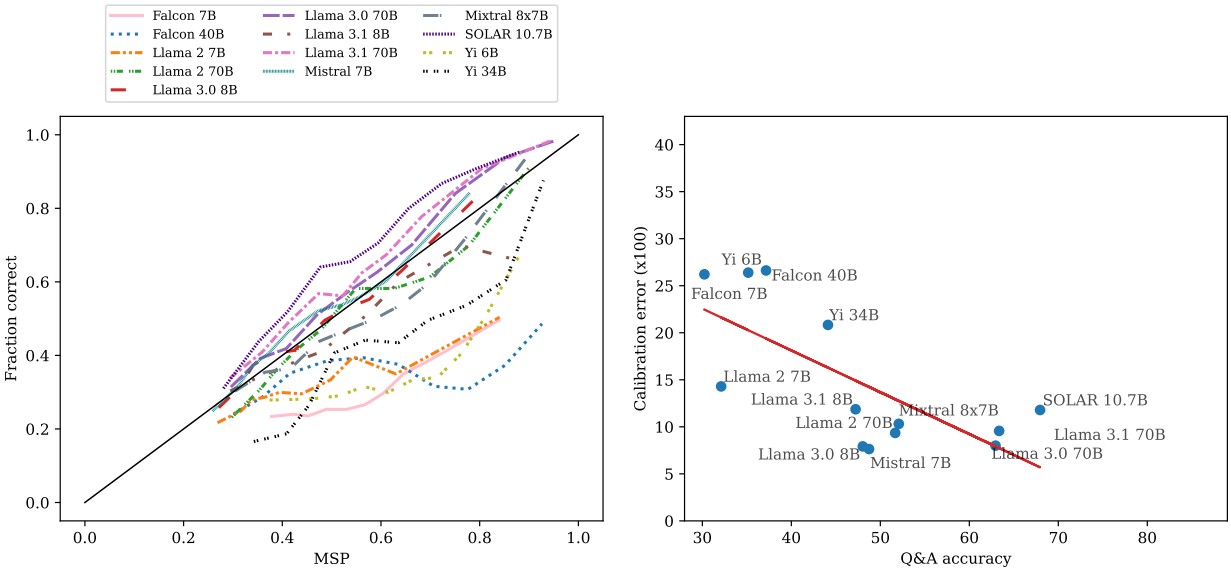

Figure 6: Calibration results for the base models. The base models are much better calibrated than the chat models, and calibration of base models *does* improve as Q&A accuracy improves ($R^2 = 0.51, p = 0.006$).

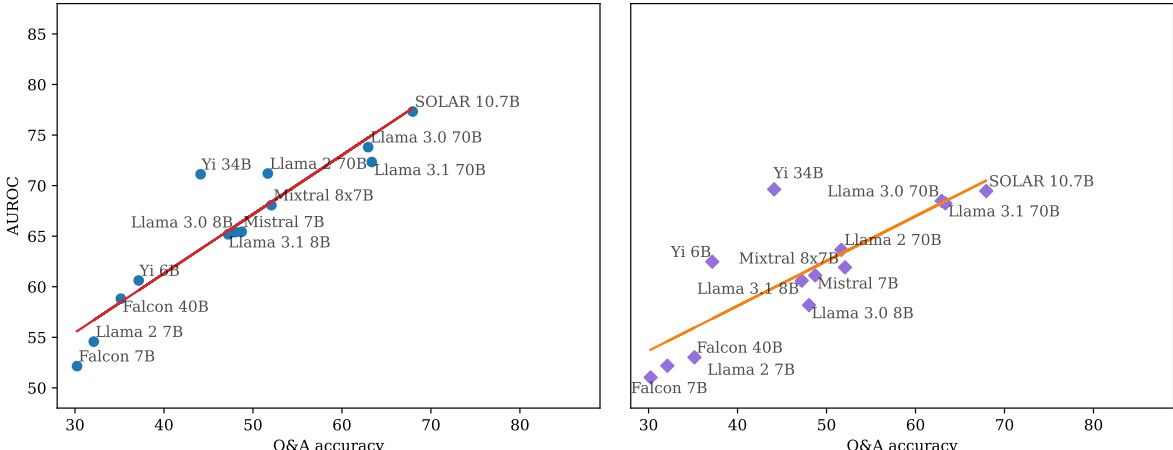

Figure 7: AUROC results for the base models for MSP (left) and Max Logit (right). We see strong correlations between Q&A accuracy and both MSP AUROC ($R^2 = 0.87$) and Max Logit AUROC ($R^2 = 0.68$), both with $p < 10^{-3}$.

**Calibration.** Figure 6 shows that base models are fairly well calibrated and that calibration further improves with Q&A accuracy ($R^2 = 0.51, p = 0.006$). Note that $p = 0.006$ does exceed our Bonferroni-adjusted significance threshold of $10^{-3}$. However, we think this is less of a concern for this result since these findings mirror those of Kadavath et al. (2022). However, it is notable that this correlation is much weaker than the correlations between AUROC and Q&A accuracy, both for chat models (Figure 3) and base models (Figure 7).

**AUROC.** Figure 7 shows the AUROC results for base models, analogous to Figure 3. We see the same strong correlation between Q&A accuracy and both MSP AUROC ($R^2 = 0.87$) and Max Logit AUROC ($R^2 = 0.68$), both with $p < 10^{-3}$. We observe that the correlations are not quite as strong as for the chat models (Figure 3), and the AUROC values themselves are somewhat lower, although this may be noise. We also note that the base models' correlations between Q&A accuracy and AUROC are much stronger than the base models' correlation between Q&A accuracy and calibration error.

## 8   Conclusion

In this paper, we provided a thorough evaluation of correctness awareness in LLMs. We showed that the MSPs of chat LLMs are miscalibrated but still provide a reliable signal of uncertainty. Furthermore, for chat LLMs, the reliability of this signal improves as model capabilities improve, but the same is not true of calibration. This contrasts with base LLMs, where correctness prediction and calibration improve in tandem. Our results pinpoint chat fine-tuning as the source of this divergence.

Our study has several limitations. One is the restriction to multiple-choice questions, which simplifies the problem in several ways. First, it removes the need to distinguish between multiple correct answers ("aleatoric uncertainty") and no good answers ("epistemic uncertainty"), both of which could result in low MSPs. The restriction to multiple-choice also enabled us to tie the LLM's answer to a single token and thus a single MSP. Future work could handle free-response questions by aggregating MSPs across tokens in a clever way. A further challenge could be multi-step decision-making problems which may involve aggregating uncertainty not only across multiple tokens in a single response, but also across multiple responses on different time steps.

Another limitation is our reliance on labeled data to transform our scientific insights into a practical method for abstention. We only used 20 data points, and labeled data was only used to choose the threshold, but a fully unsupervised method would be advantageous in many settings.

We would also like to better understand when and why these methods fail. Are there particular subcategories of unfamiliar situations that are especially challenging to identify? For example, why was the WinoGrande dataset so much harder for our correctness prediction task?

More broadly, we are excited about developing more robust methods for mistake detection in LLMs, both for Q&A tasks and other contexts.

**Broader impact statement**

The capabilities of AI systems have advanced rapidly over the past several years and will likely continue to grow. In order to ensure that AI is beneficial for society, we believe it is paramount to understand and mitigate the risks of such systems. In this paper, we focus on one particular risk: harmful responses from LLMs. We hope that our work contributes to the ongoing efforts to mitigate harmful LLM responses.

We do not think it is likely for our work to inadvertently cause harm, but one possibility is worth mentioning. If a reader were to assume that abstention methods can completely eliminate false responses, that reader might be more likely to fall prey to false responses when they do inevitably still occur. We caution readers to remain vigilant about false responses from LLMs.

**Acknowledgements**

This work was supported in part by a gift from Open Philanthropy to the Center for Human-Compatible AI (CHAI) at UC Berkeley. We would like to thank (in alphabetical order) Cassidy Laidlaw, Katie Kang, Peter Hase, and Yaodong Yu for helpful discussions and feedback.

**Author contributions**

B. Plaut conceived the project with feedback from N. Khanh and T. Trinh. B. Plaut led the project. B. Plaut designed the initial correctness prediction experiments. N. Khanh suggested adding calibration analysis. All authors contributed to experiment design. B. Plaut implemented and ran all experiments. T. Trinh implemented and ran the statistical analysis. B. Plaut implemented and ran the overall analysis and visualization pipeline. B. Plaut led the writing, with significant contributions from N. Khanh and T. Trinh.

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

```
You will be asked a multiple-choice question.  Respond with the letter which corresponds
to the correct answer, followed by a period.  There is no need to provide an explanation,
so your response should be very short.  Now here is the question:

In the nitrogen cycle, nitrogen can return to the lithosphere directly from the atmosphere
by
A. lightning.
B. cellular respiration.
C. air pollution.
D. condensation.

Answer:
```

Figure 8: The second prompt phrasing we used.

## A    Details on main experiments

Here we include details on the methodology and results of our main experiments. First, we include the second prompt phrasing we used (Figure 8). Recall that Figure 1 shows the first phrasing.

### A.1    Details on calibration results

Table 6: Calibration error and Q&A accuracy for each model.

| LLM | Q&A accuracy | Calibration error ($\times 100$) |
|---|---|---|
| Falcon 7B | 30.1 | 14.8 |
| Falcon 40B | 40.5 | 11.9 |
| Llama 2 7B | 39.7 | 40.7 |
| Llama 2 70B | 58.7 | 32.8 |
| Llama 3.0 8B | 59.1 | 20.4 |
| Llama 3.0 70B | 78.8 | 16.5 |
| Llama 3.1 8B | 62.0 | 10.8 |
| Llama 3.1 70B | 80.6 | 9.2 |
| Mistral 7B | 57.0 | 40.9 |
| Mixtral 8x7B | 69.9 | 29.6 |
| SOLAR 10.7B | 67.5 | 30.8 |
| Yi 6B | 49.6 | 27.3 |
| Yi 34B | 69.1 | 17.4 |
| GPT-3.5 Turbo | 67.5 | 21.5 |
| GPT-4o | 86.9 | 11.5 |

As discussed in Section 4, each model's calibration error was computed by first computing the calibration error for each combination of model-dataset-phrasing, and then averaging over datasets and phrasings to obtain a per-model total error. This was done to avoid downweighting datasets with fewer questions. This data is shown in Figure 2 (right) in Section 4 and also in Table 6 here in this appendix.

However, this approach does not make sense for the calibration curves in Figure 2. This is because calibration curves are not averages: they are obtained by bucketing each data point (in our case, a question-response pair is a data point). In order to avoid downweighting the smaller datasets, we duplicated data points from those datasets so that the total number of points per dataset was 6,000. Concretely, we duplicated each data

point from TruthfulQA by 7 to obtain $7 \cdot 817 = 5719$ questions and then randomly sampled an additional $6,000 - 5,719$ questions to obtain 6,000 total questions. Similarly, we duplicated each data point from ARC by 2, and then randomly sampled $6,000 - 2 \cdot 2,590$ extra questions. Although this solution has drawbacks, we felt that it was the most reasonable option.

## A.2 Details on Q&A-with-abstention results

Figure 9 is based on the same experimental data as Table 4, but shows each model's performance across the entire range of possible thresholds. A threshold of zero corresponds to the base LLM and the black dot indicates the threshold chosen during the training phase (using 20 labeled data points per dataset), which is also the threshold used to compute the score in Table 4. One can see that the chosen thresholds are not quite optimal, but 20 data points was still enough to produce substantial improvements over the baseline of not abstaining.

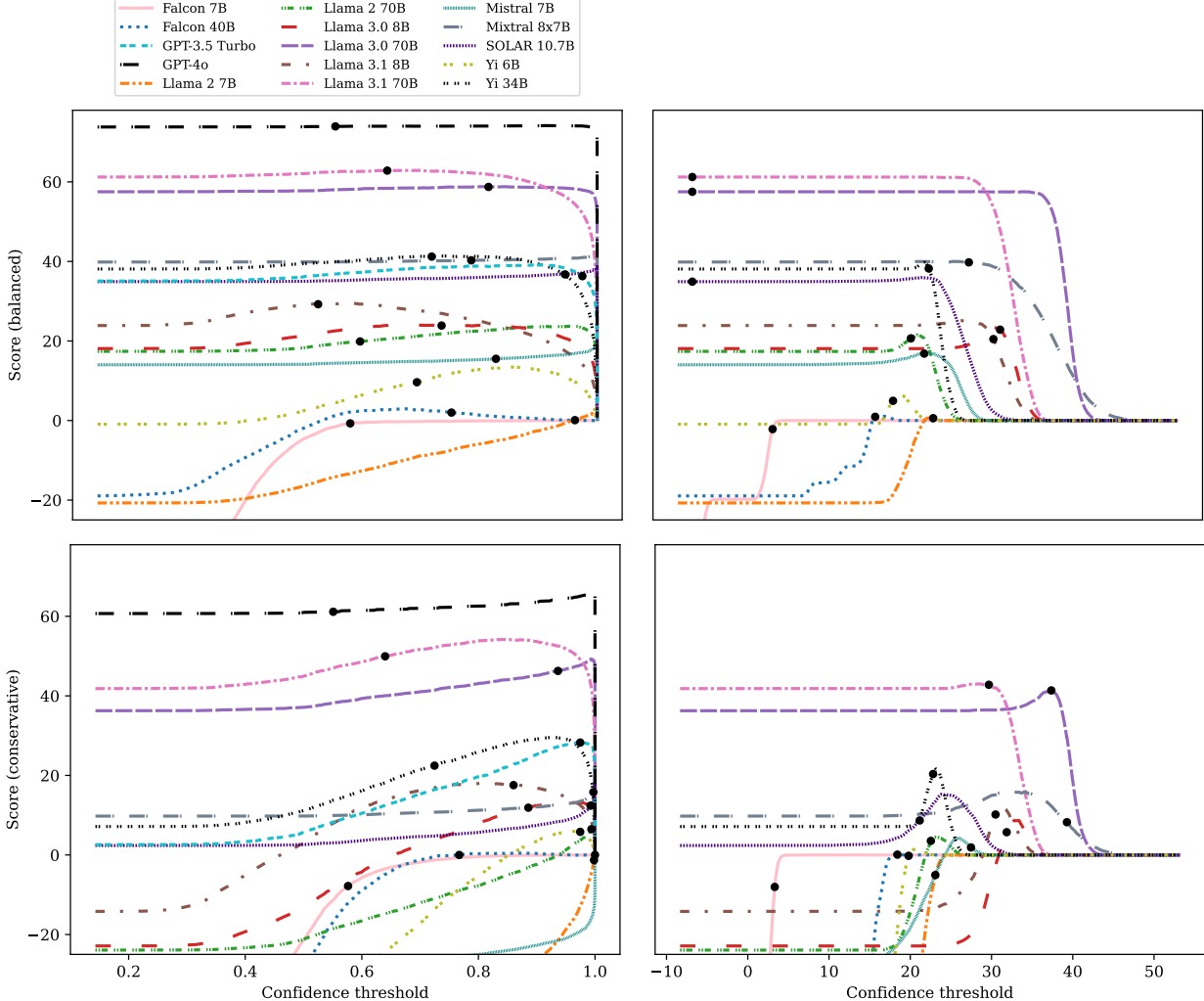

Figure 9: Q&A-with-abstention scores across all possible thresholds. The base LLM corresponds to a threshold of zero. The black dot indicates the threshold selected via the training set, which determines the MSP and Max Logit scores in Table 4.

We can also visualize this by plotting the performance of each model as a function of the amount of training data, as shown in Figure 10.

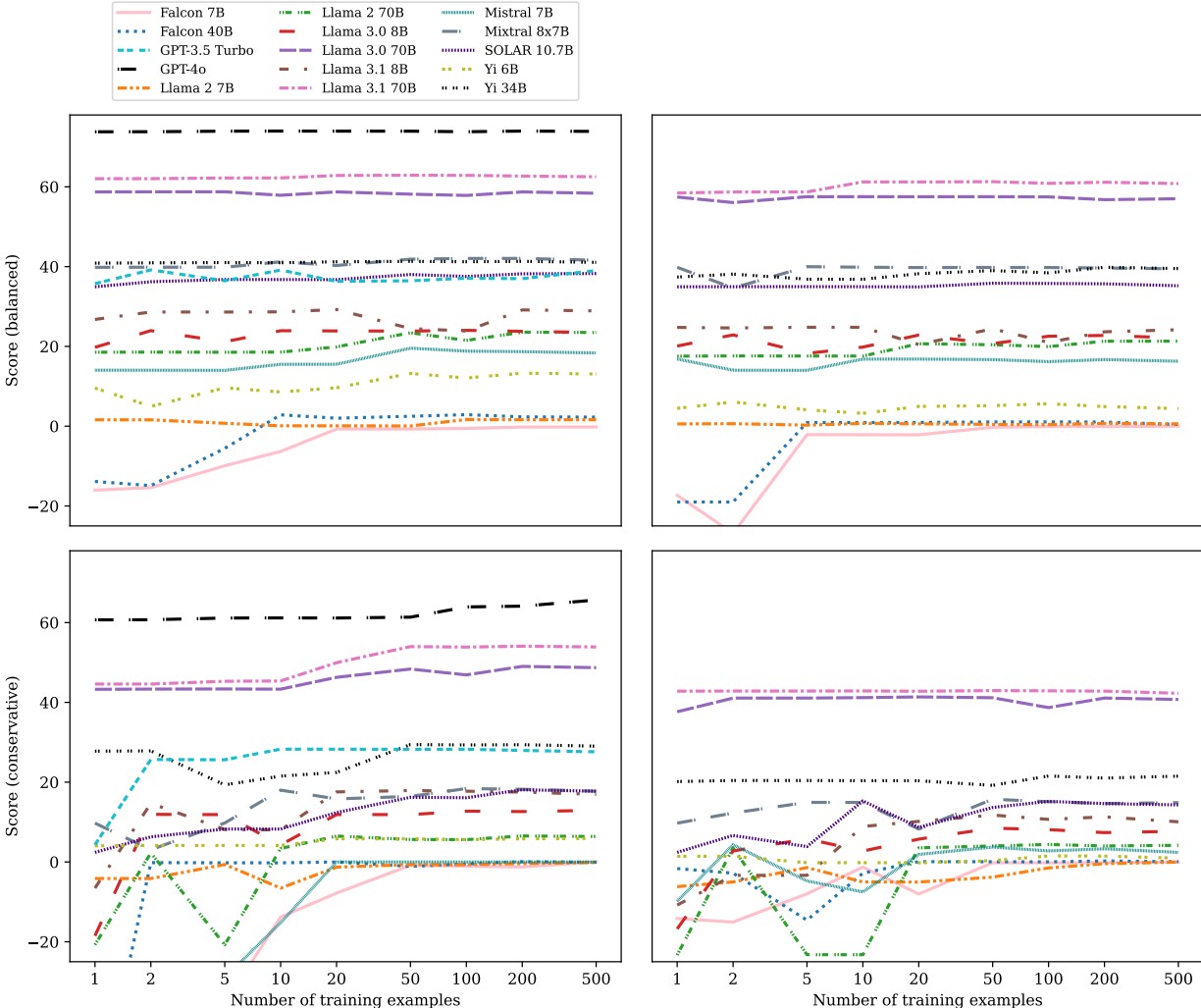

Figure 10: Q&A-with-abstention scores as a function of the amount of training data. The x-axis is the number of data points per dataset included in the training data (referred to as $k$ in Section 6.1). Overall, scores significantly improve as $k$ increases to about 20, but there is minimal improvement if $k$ increases beyond that.

Table 7 shows the average abstention frequencies across all datasets, corresponding to the scores from Table 4.

Table 7: Frequency of abstention in the Section 6 experiments, as percentages.

| LLM | Balanced | | | Conservative | | |
|---|---|---|---|---|---|---|
| | No abstain | MSP | Max Logit | No abstain | MSP | Max Logit |
| Falcon 7B | 0 | 86.5 | 90.4 | 0 | 86.5 | 90.4 |
| Falcon 40B | 0 | 93.7 | 68.1 | 0 | 94.7 | 99.7 |
| Llama 2 7B | 0 | 66.9 | 88.1 | 0 | 94.4 | 88.1 |
| Llama 2 70B | 0 | 6.7 | 14.2 | 0 | 50.1 | 52.2 |
| Llama 3.0 8B | 0 | 36.8 | 36.9 | 0 | 54.7 | 47.6 |
| Llama 3.0 70B | 0 | 10.4 | 0.0 | 0 | 17.1 | 19.4 |
| Llama 3.1 8B | 0 | 22.9 | 58.9 | 0 | 67.2 | 58.9 |
| Llama 3.1 70B | 0 | 11.4 | 0.0 | 0 | 11.4 | 13.8 |
| Mistral 7B | 0 | 4.5 | 29.6 | 0 | 100.0 | 92.7 |
| Mixtral 8x7B | 0 | 1.4 | 6.2 | 0 | 6.7 | 71.1 |
| SOLAR 10.7B | 0 | 6.8 | 0.0 | 0 | 11.5 | 9.7 |
| Yi 6B | 0 | 34.3 | 36.6 | 0 | 86.8 | 83.0 |
| Yi 34B | 0 | 20.4 | 22.9 | 0 | 21.1 | 30.3 |
| GPT-3.5 Turbo | 0 | 47.6 | | 0 | 47.6 | |
| GPT-4o | 0 | 0.4 | | 0 | 0.4 | |

### A.3 Details on base models

Table 8 provides the precise AUROC and Q&A accuracy values in Figure 7 (analogous to Figure 3). Table 9 provides Q&A-with-abstention results for the base models (analogous to Table 4). As with the chat models, we see that $k = 20$ labeled data points per dataset is sufficient to improve over the baseline of never abstaining.

Table 8: AUROC results for the base models. See Table 2 for an explanation of the $p$-values.

| LLM | Q&A accuracy | MSP AUROC | MSP $p < 10^{-4}$ | Max Logit AUROC | Max Logit $p < 10^{-4}$ |
|---|---|---|---|---|---|
| Falcon 7B | 30.2 | 52.1 | 1/10 | 51.0 | 0/10 |
| Falcon 40B | 35.2 | 58.8 | 7/10 | 53.0 | 2/10 |
| Llama 2 7B | 32.1 | 54.6 | 3/10 | 52.2 | 1/10 |
| Llama 2 70B | 51.7 | 71.2 | 10/10 | 63.7 | 8/10 |
| Llama 3.0 8B | 48.0 | 65.4 | 8/10 | 58.2 | 6/10 |
| Llama 3.0 70B | 62.9 | 73.8 | 10/10 | 68.5 | 10/10 |
| Llama 3.1 8B | 47.2 | 65.2 | 9/10 | 60.6 | 7/10 |
| Llama 3.1 70B | 63.4 | 72.3 | 9/10 | 68.2 | 10/10 |
| Mistral 7B | 48.7 | 65.4 | 7/10 | 61.1 | 8/10 |
| Mixtral 8x7B | 52.1 | 68.1 | 10/10 | 61.9 | 8/10 |
| SOLAR 10.7B | 68.0 | 77.3 | 10/10 | 69.5 | 10/10 |
| Yi 6B | 37.2 | 60.6 | 9/10 | 62.5 | 10/10 |
| Yi 34B | 44.1 | 71.1 | 10/10 | 69.6 | 10/10 |

Table 9: Q&A-with-abstention results for the base models. We see roughly the same patterns as in the Q&A-with-abstention results for the chat models (Figure 9). See Table 4 for an explanation of the scoring scheme.

| LLM | Balanced No abstain | Balanced MSP | Balanced Max Logit | Conservative No abstain | Conservative MSP | Conservative Max Logit |
|---|---|---|---|---|---|---|
| Falcon 7B | −39.6 | 0.1 | −0.4 | −109.3 | −0.5 | −1.3 |
| Falcon 40B | −29.6 | 0.1 | −1.1 | −94.4 | −0.4 | 0.0 |
| Llama 2 7B | −35.7 | −2.7 | −0.1 | −103.5 | −0.7 | −0.3 |
| Llama 2 70B | 3.4 | 15.6 | 12.1 | −44.9 | 8.1 | 5.2 |
| Llama 3.0 8B | −4.0 | 8.5 | 0.3 | −56.0 | 5.3 | 1.1 |
| Llama 3.0 70B | 25.9 | 33.3 | 30.8 | −11.2 | 23.0 | 15.6 |
| Llama 3.1 8B | −5.7 | 9.3 | 5.6 | −58.5 | −3.8 | 1.3 |
| Llama 3.1 70B | 26.6 | 26.6 | 30.9 | −10.1 | 4.5 | 4.0 |
| Mistral 7B | −2.5 | 10.7 | 6.7 | −53.7 | 3.3 | 0.2 |
| Mixtral 8x7B | 4.2 | 15.3 | 9.2 | −43.6 | 9.0 | 3.0 |
| SOLAR 10.7B | 35.9 | 40.9 | 36.0 | 3.8 | 22.5 | 15.8 |
| Yi 6B | −25.7 | 3.2 | 1.3 | −88.5 | −0.9 | −0.4 |
| Yi 34B | −11.7 | 10.2 | 11.4 | −67.5 | 6.2 | 9.2 |

# B Results for 5-shot prompting

As discussed in Section 3, our main experiments used zero-shot prompting in order to test our hypothesis in the simplest setting possible. In this section, we show how the results change if 5-shot prompting is used instead. In short, the same correlations and trends hold for MSP (although not Max Logit). The AUROC values are also slightly lower overall, which could suggest that few-shot prompting disrupts the innate uncertainty information in LLMs, but we do not have the evidence to conclude this definitively. Our 5-shot experiments used the same setup as our zero-shot experiments, simply with 5 examples prepended to each prompt.

## B.1 Chat models

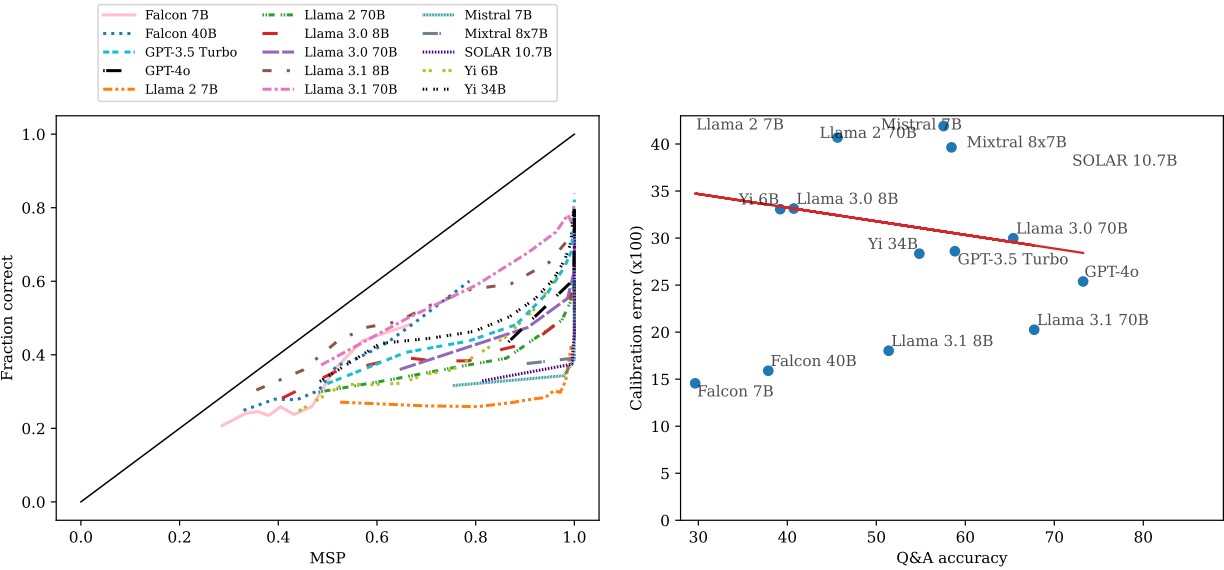

Figure 11: Calibration results for 5-shot prompting for chat models. Calibration is somewhat worse than for zero-shot prompting with chat models (Figure 2), but this may be noise. Similar to zero-shot prompting, there is no statistical evidence for a correlation between calibration error and Q&A accuracy ($R^2 = 0.03, p = 0.57$).

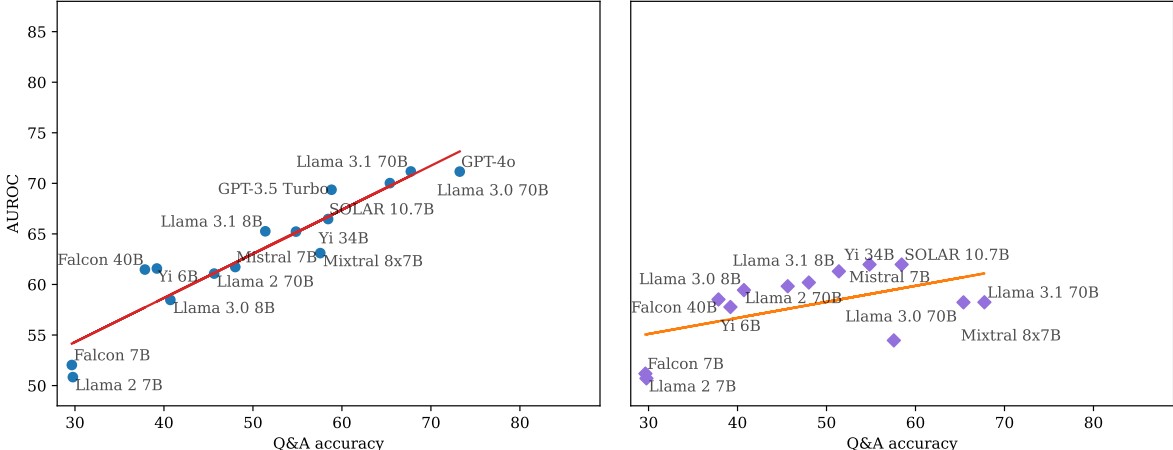

Figure 12: AUROC results for 5-shot prompting for chat models. AUROC values are worse compared to zero-shot, but this may be noise. MSP AUROC (left) retains a strong correlation with Q&A accuracy ($R^2 = 0.88, p < 10^{-4}$), but the correlation with Max Logit (right) is much weaker here ($R^2 = 0.28, p = 0.06$).

Table 10: AUROC results for 5-shot prompting for chat models. See Table 2 for an explanation of the $p$-values.

| | | MSP | | Max Logit | |
|---|---|---|---|---|---|
| LLM | Q&A accuracy | AUROC | $p < 10^{-4}$ | AUROC | $p < 10^{-4}$ |
| Falcon 7B | 29.6 | 52.0 | 1/10 | 51.2 | 0/10 |
| Falcon 40B | 37.9 | 61.5 | 10/10 | 58.5 | 8/10 |
| Llama 2 7B | 29.8 | 50.8 | 0/10 | 50.7 | 0/10 |
| Llama 2 70B | 45.6 | 61.1 | 8/10 | 59.8 | 8/10 |
| Llama 3.0 8B | 40.7 | 58.5 | 8/10 | 59.4 | 7/10 |
| Llama 3.0 70B | 65.4 | 70.0 | 10/10 | 58.2 | 7/10 |
| Llama 3.1 8B | 51.4 | 65.3 | 10/10 | 61.3 | 10/10 |
| Llama 3.1 70B | 67.7 | 71.2 | 10/10 | 58.2 | 7/10 |
| Mistral 7B | 48.0 | 61.7 | 10/10 | 60.2 | 9/10 |
| Mixtral 8x7B | 57.6 | 63.1 | 10/10 | 54.5 | 4/10 |
| SOLAR 10.7B | 58.4 | 66.5 | 10/10 | 62.0 | 10/10 |
| Yi 6B | 39.2 | 61.6 | 9/10 | 57.8 | 6/10 |
| Yi 34B | 54.8 | 65.2 | 10/10 | 62.0 | 10/10 |
| GPT-3.5 Turbo | 58.8 | 69.4 | 10/10 | | |
| GPT-4o | 73.2 | 71.2 | 10/10 | | |

Table 11: Q&A-with-abstention results for 5-shot prompting for chat models. See Table 4 for an explanation of the scoring scheme.

| | Balanced | | | Conservative | | |
|---|---|---|---|---|---|---|
| LLM | No abstain | MSP | Max Logit | No abstain | MSP | Max Logit |
| Falcon 7B | −40.7 | −1.8 | −0.9 | −111.1 | −1.3 | −2.6 |
| Falcon 40B | −24.3 | 1.4 | 0.3 | −86.5 | −1.2 | −0.2 |
| Llama 2 7B | −40.4 | −0.7 | 0.0 | −110.6 | −1.7 | −1.1 |
| Llama 2 70B | −8.7 | 4.0 | 2.3 | −63.1 | 0.0 | 0.0 |
| Llama 3.0 8B | −18.7 | 0.6 | 0.9 | −78.0 | −0.9 | −0.1 |
| Llama 3.0 70B | 30.8 | 33.4 | 31.4 | −3.8 | 15.6 | 6.1 |
| Llama 3.1 8B | 2.8 | 11.2 | 8.6 | −45.8 | 1.1 | 1.2 |
| Llama 3.1 70B | 35.5 | 37.5 | 35.7 | 3.3 | 18.3 | 12.6 |
| Mistral 7B | −4.0 | 2.6 | 1.8 | −56.0 | 0.0 | 0.1 |
| Mixtral 8x7B | 15.1 | 15.2 | 15.9 | −27.3 | 1.4 | −4.1 |
| SOLAR 10.7B | 16.9 | 21.3 | 21.2 | −24.7 | 0.0 | 1.6 |
| Yi 6B | −21.6 | 2.4 | 0.5 | −82.4 | 0.4 | −0.5 |
| Yi 34B | 9.6 | 9.7 | 12.7 | −35.6 | 2.6 | 3.1 |
| GPT-3.5 Turbo | 17.6 | 20.5 | | −23.5 | 10.6 | |
| GPT-4o | 46.6 | 47.8 | | 19.9 | 28.5 | |

## B.2 Base models

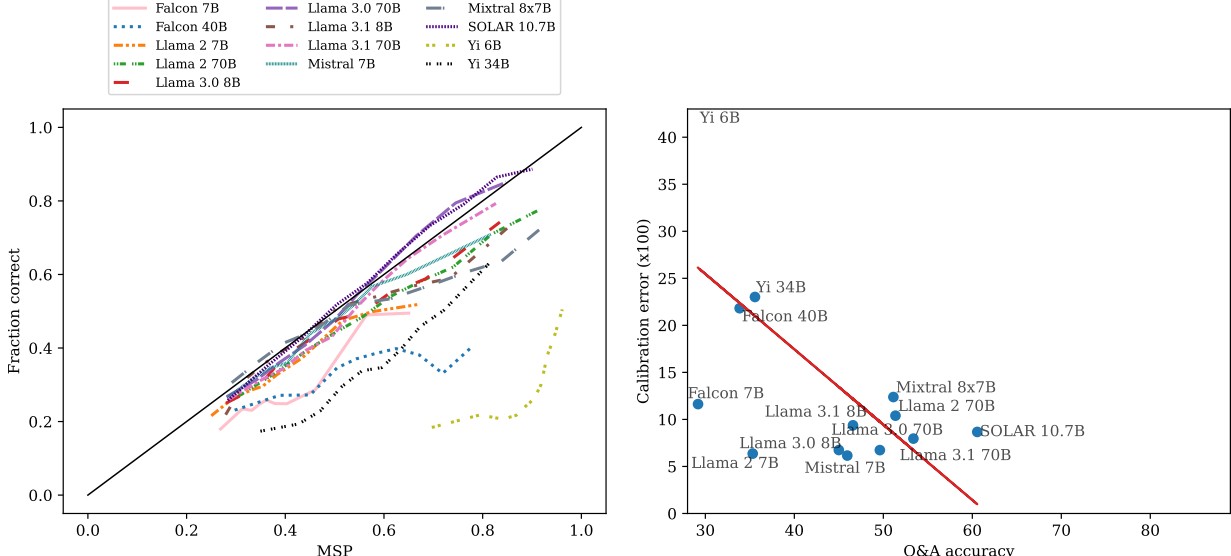

Figure 13: Calibration results for 5-shot prompting for base models. The results are similar to the zero-shot prompting results for base models, although the correlation between calibration error and Q&A accuracy is weaker here ($R^2 = 0.32, p = 0.04$).

Table 12: AUROC results for 5-shot prompting for base models. See Table 2 for an explanation of the $p$-values.

| LLM | Q&A accuracy | MSP | | Max Logit | |
|---|---|---|---|---|---|
| | | AUROC | $p < 10^{-4}$ | AUROC | $p < 10^{-4}$ |
| Falcon 7B | 29.2 | 52.4 | 2/10 | 51.8 | 1/10 |
| Falcon 40B | 33.9 | 57.5 | 8/10 | 57.1 | 5/10 |
| Llama 2 7B | 35.3 | 57.3 | 6/10 | 53.2 | 4/10 |
| Llama 2 70B | 51.4 | 65.9 | 9/10 | 63.8 | 9/10 |
| Llama 3.0 8B | 45.0 | 62.7 | 6/10 | 59.3 | 6/10 |
| Llama 3.0 70B | 53.4 | 66.3 | 10/10 | 63.0 | 10/10 |
| Llama 3.1 8B | 46.6 | 65.4 | 9/10 | 60.9 | 8/10 |
| Llama 3.1 70B | 49.6 | 64.5 | 10/10 | 61.9 | 10/10 |
| Mistral 7B | 46.0 | 62.5 | 8/10 | 58.3 | 6/10 |
| Mixtral 8x7B | 51.1 | 64.5 | 10/10 | 56.2 | 6/10 |
| SOLAR 10.7B | 60.6 | 70.8 | 10/10 | 64.4 | 10/10 |
| Yi 6B | 29.4 | 59.2 | 8/10 | 53.9 | 4/10 |
| Yi 34B | 35.6 | 65.5 | 10/10 | 60.7 | 9/10 |

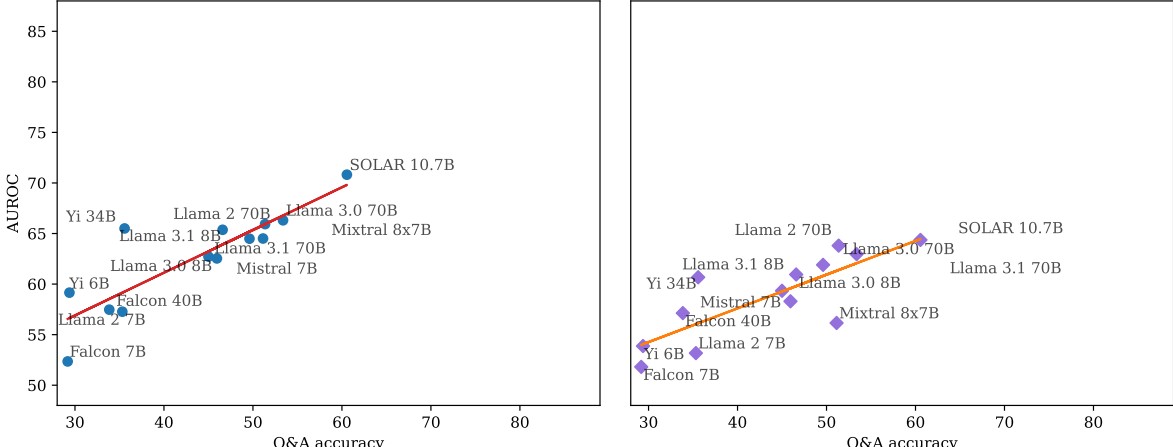

Figure 14: AUROC results for 5-shot prompting for base models. The results are similar to zero-shot results for base models. Here the coefficients of determination with Q&A accuracy are $R^2 = 0.74$ and $R^2 = 0.64$ for MSP (left) and Max Logit (right) AUROC respectively, both with $p < 10^{-3}$.

Table 13: Q&A-with-abstention results for 5-shot prompting for base models. See Table 4 for an explanation of the scoring scheme.

| | Balanced | | | Conservative | | |
|---|---|---|---|---|---|---|
| LLM | No abstain | MSP | Max Logit | No abstain | MSP | Max Logit |
| Falcon 7B | −41.8 | 0.0 | −3.2 | −112.6 | −1.3 | −0.2 |
| Falcon 40B | −32.3 | −1.8 | −4.5 | −98.5 | −7.5 | −2.4 |
| Llama 2 7B | −29.4 | 0.2 | 0.0 | −94.0 | −0.7 | −0.2 |
| Llama 2 70B | 2.7 | 12.7 | 11.4 | −45.9 | 3.9 | −1.3 |
| Llama 3.0 8B | −10.1 | 7.2 | 4.4 | −65.1 | −2.7 | 0.1 |
| Llama 3.0 70B | 6.9 | 17.7 | 19.4 | −39.7 | 7.5 | 5.9 |
| Llama 3.1 8B | −6.8 | 8.0 | 4.7 | −60.2 | 1.7 | −1.9 |
| Llama 3.1 70B | −0.7 | 14.4 | 10.5 | −51.1 | 4.0 | 1.4 |
| Mistral 7B | −8.1 | 5.3 | 3.2 | −62.1 | 0.8 | 0.1 |
| Mixtral 8x7B | 2.3 | 11.7 | 7.6 | −46.6 | 1.5 | 1.2 |
| SOLAR 10.7B | 21.1 | 29.7 | 23.9 | −18.4 | 18.0 | 9.5 |
| Yi 6B | −41.2 | 0.1 | −0.6 | −111.9 | −0.8 | −1.8 |
| Yi 34B | −28.8 | 1.8 | 0.8 | −93.3 | −3.7 | −1.1 |

## C    Other measures of uncertainty

Up until this point, we have only considered the maximum softmax probability and the maximum logit. However, there are other ways to measure the uncertainty of a distribution. Here we consider margin (the difference between the maximum probability/logit and the second largest probability/logit) and entropy (only for probabilities). All results in this section use chat models and zero-shot prompting. We omit calibration analysis for the margin and entropy metrics, because calibration relates to the probability of a particular outcome, and margin and entropy do not correspond to probabilities of any outcome.

Table 14: AUROC results using the entropy of the softmax probabilities instead of the MSP. See Table 2 for explanation of the $p$-values.

| | | Entropy | |
| LLM | Q&A accuracy | AUROC | $p < 10^{-4}$ |
| --- | --- | --- | --- |
| Falcon 7B | 30.1 | 52.8 | 1/10 |
| Falcon 40B | 40.5 | 60.6 | 7/10 |
| Llama 2 7B | 39.7 | 58.9 | 6/10 |
| Llama 2 70B | 58.7 | 71.1 | 10/10 |
| Llama 3.0 8B | 59.1 | 71.0 | 10/10 |
| Llama 3.0 70B | 78.8 | 81.3 | 10/10 |
| Llama 3.1 8B | 62.0 | 72.3 | 10/10 |
| Llama 3.1 70B | 80.6 | 83.3 | 10/10 |
| Mistral 7B | 57.0 | 66.1 | 10/10 |
| Mixtral 8x7B | 69.9 | 70.0 | 10/10 |
| SOLAR 10.7B | 67.5 | 71.8 | 10/10 |
| Yi 6B | 49.6 | 68.5 | 10/10 |
| Yi 34B | 69.1 | 75.9 | 10/10 |
| GPT-3.5 Turbo | 67.5 | 76.2 | 10/10 |
| GPT-4o | 86.9 | 85.3 | 10/10 |

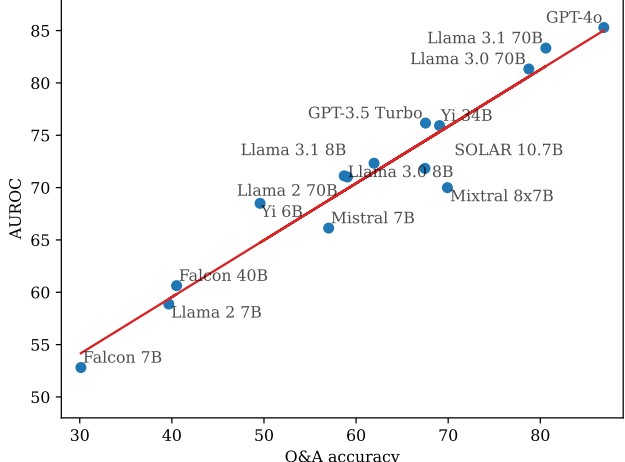

Figure 15: AUROC results using the entropy of the probability distribution as the uncertainty metric. The AUROC values are similar to those obtained from the MSP, and the entropy AUROC also exhibits a strong correlation with Q&A accuracy ($R^2 = 0.93, p < 10^{-4}$).

Table 15: AUROC results using the margin of softmax probabilities and margin of logits instead of the MSP and maximum logit, respectively. See Table 2 for explanation of the $p$-values.

| LLM | Q&A accuracy | Probability margin | | Logit margin | |
|---|---|---|---|---|---|
| | | AUROC | $p < 10^{-4}$ | AUROC | $p < 10^{-4}$ |
| Falcon 7B | 30.1 | 50.1 | 0/10 | 49.6 | 0/10 |
| Falcon 40B | 40.5 | 59.2 | 5/10 | 58.7 | 5/10 |
| Llama 2 7B | 39.7 | 58.3 | 6/10 | 58.1 | 6/10 |
| Llama 2 70B | 58.7 | 71.0 | 10/10 | 71.0 | 10/10 |
| Llama 3.0 8B | 59.1 | 71.1 | 10/10 | 70.9 | 10/10 |
| Llama 3.0 70B | 78.8 | 81.3 | 10/10 | 81.3 | 10/10 |
| Llama 3.1 8B | 62.0 | 72.9 | 10/10 | 72.7 | 10/10 |
| Llama 3.1 70B | 80.6 | 83.4 | 10/10 | 83.4 | 10/10 |
| Mistral 7B | 57.0 | 66.2 | 10/10 | 66.2 | 10/10 |
| Mixtral 8x7B | 69.9 | 70.0 | 10/10 | 70.1 | 10/10 |
| SOLAR 10.7B | 67.5 | 71.8 | 10/10 | 71.9 | 10/10 |
| Yi 6B | 49.6 | 66.0 | 10/10 | 65.4 | 10/10 |
| Yi 34B | 69.1 | 75.6 | 10/10 | 75.5 | 10/10 |
| GPT-3.5 Turbo | 67.5 | 76.0 | 10/10 | | |
| GPT-4o | 86.9 | 85.3 | 10/10 | | |

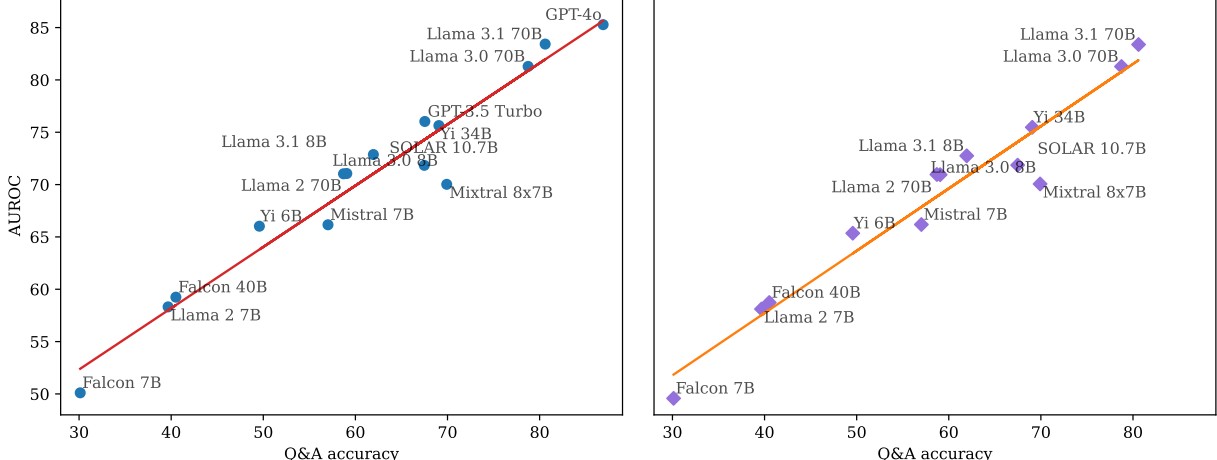

Figure 16: AUROC results using margin (difference between two highest answer probabilities or answer logits) as the uncertainty metric instead of maximum. **Left:** probability margin AUROC values. These AUROC values and the correlation strength with Q&A accuracy ($R^2 = 0.95, p < 10^{-4}$) are similar to MSP. **Right:** logit margin AUROC values. These AUROC values and the correlation strength with Q&A accuracy ($R^2 = 0.94, p < 10^{-4}$) for logit margin are significantly higher than for the maximum logit. Overall, for both probabilities and logits, the same qualitative trends hold, whether the margin or the maximum is used.

# D Post-hoc calibration

As discussed in Section 2, there exist many methods for rescaling the MSPs of a miscalibrated classifier to improve calibration. One method is Platt scaling (Platt, 2000). Given an MSP $x$, the Platt-scaled MSP is

$$\text{Platt}_{A,B}(x) = \frac{1}{1 + \exp(Ax + B)}$$

where $A$ and $B$ are real-valued parameters learned from data. For training data, we use the same set of $k = 20$ data points per dataset as in the Q&A-with-abstention experiments (Section 6). For each model, we aggregate all the training data across datasets and optimize $A$ and $B$ across the resulting dataset. Figure 17 and Table 16 present calibration analysis for the Platt-scaled MSPs. Compared to Figure 2, calibration error is much less. However, there remains no correlation between Q&A accuracy and calibration error ($R^2 = 0.13, p = 0.18$).

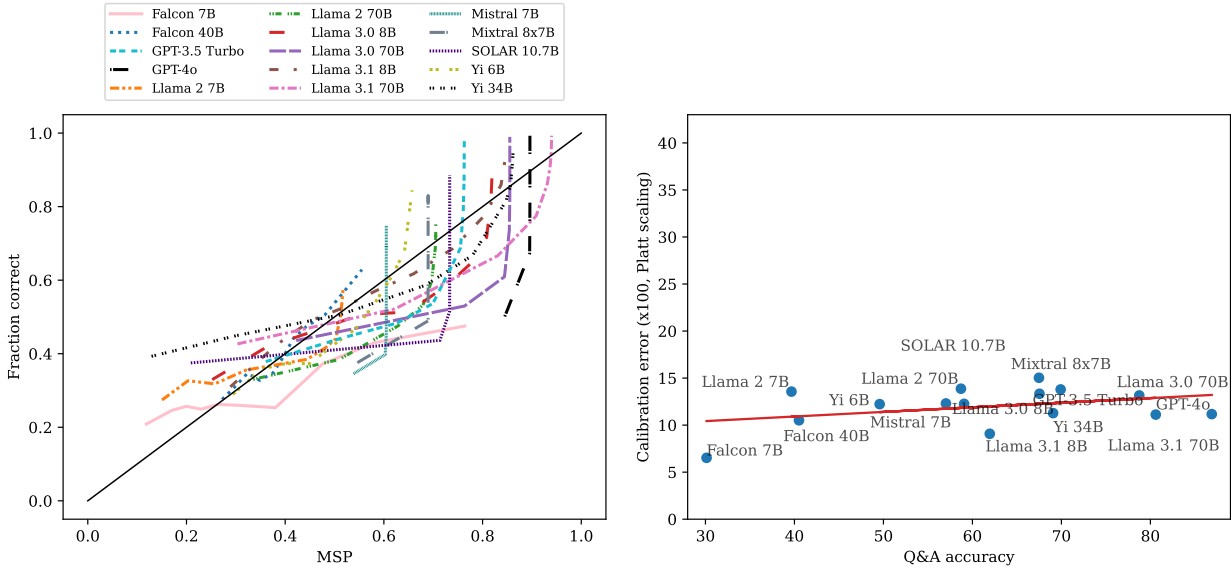

Figure 17: Calibration analysis for Platt-scaled MSPs (for chat models and zero-shot prompting). In the left plot, $A$ ranges from 1.4 to 29.3 with a median of 4.1, while $B$ ranges from -28.3 to -0.5 with a median of -3.1. (The right plot has different values of $A$ and $B$ since as before, we computes calibration error separately for each combination of model, prompt phrasing, and dataset and then average to obtain the calibration error per model.)

We also explored other parameters for $A$ and $B$. For some choices of $A$ and $B$, we can recover the correlation between calibration error and Q&A accuracy. However, this correlation reverses for other choices of $A$ and $B$. It is perhaps unsurprising that we can obtain arbitrary correlations by manually tweaking these parameters.

Table 16: Calibration error after Platt scaling and Q&A accuracy for each chat model.

| LLM | Q&A accuracy | Calibration error ($\times 100$) |
|---|---|---|
| Falcon 7B | 30.1 | 6.5 |
| Falcon 40B | 40.5 | 10.5 |
| Llama 2 7B | 39.7 | 13.6 |
| Llama 2 70B | 58.7 | 13.9 |
| Llama 3.0 8B | 59.1 | 12.3 |
| Llama 3.0 70B | 78.8 | 13.2 |
| Llama 3.1 8B | 62.0 | 9.1 |
| Llama 3.1 70B | 80.6 | 11.1 |
| Mistral 7B | 57.0 | 12.3 |
| Mixtral 8x7B | 69.9 | 13.8 |
| SOLAR 10.7B | 67.5 | 15.0 |
| Yi 6B | 49.6 | 12.2 |
| Yi 34B | 69.1 | 11.3 |
| GPT-3.5 Turbo | 67.5 | 13.3 |
| GPT-4o | 86.9 | 11.2 |

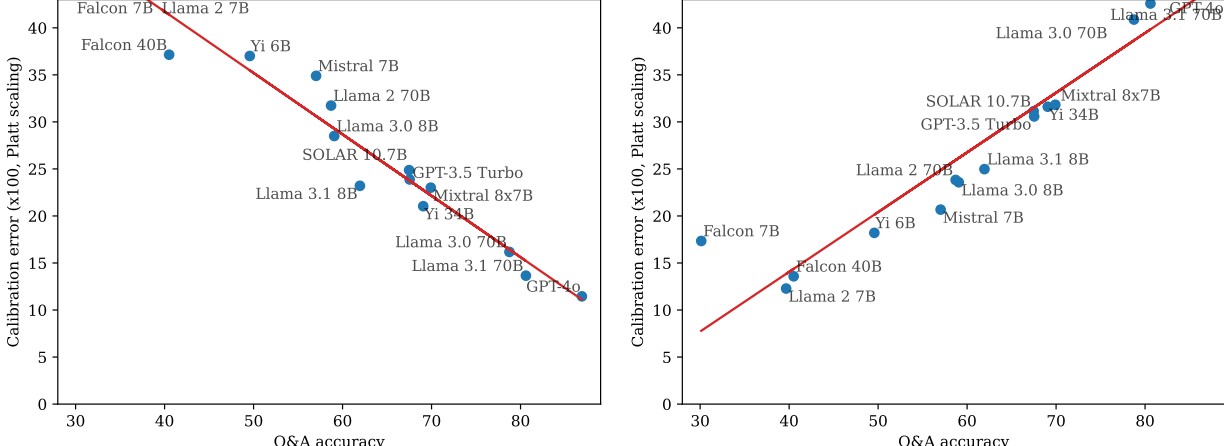

Figure 18: Calibration analysis for Platt-scaled MSPs with alternative $A, B$ parameters (for chat models and zero-shot prompting). When we explicitly set $A = 2.5$ and $B = 0$ (left), we recover a strong negative correlation between calibration error and Q&A accuracy ($R^2 = 0.93, p < 10^{-4}$). However, for $A = -0.5$ and $B = 0$ (right), this correlation reverses ($R^2 = 0.89, p < 10^{-4}$). In both cases, the calibration errors themselves are much larger overall than when $A, B$ are chosen by optimizing over the training data (Figure 17).

# E    Dataset-level results

This section presents per-dataset versions of Figure 2 (calibration), Table 2 and Figure 3 (AUROC), and Table 4 and Figure 10 (Q&A with abstention).

## E.1    ARC-Challenge

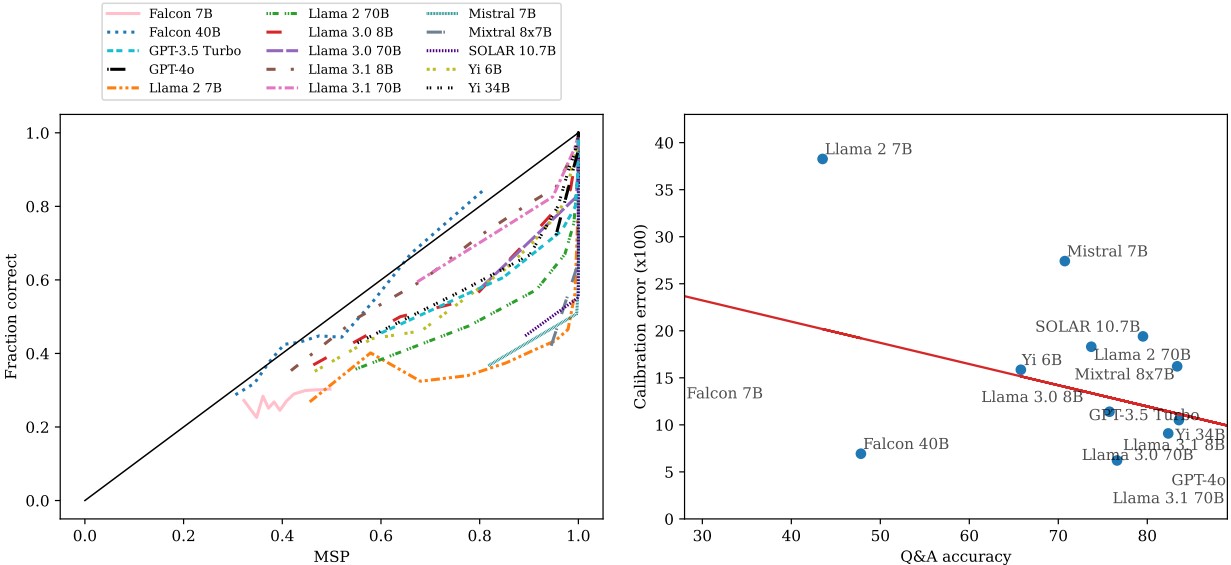

Figure 19: Calibration results for ARC-Challenge. The coefficient of determination between Q&A accuracy and calibration error was $R^2 = 0.21$ ($p = 0.08$).

Table 17: AUROC results for ARC-Challenge. See Table 2 for explanation of the $p$-values.

| | | MSP | | Max Logit | |
| LLM | Q&A accuracy | AUROC | $p < 10^{-4}$ | AUROC | $p < 10^{-4}$ |
|---|---|---|---|---|---|
| Falcon 7B | 27.1 | 52.1 | 0/2 | 51.3 | 0/2 |
| Falcon 40B | 47.8 | 68.7 | 2/2 | 67.1 | 2/2 |
| Llama 2 7B | 43.5 | 63.6 | 2/2 | 62.2 | 2/2 |
| Llama 2 70B | 73.7 | 78.7 | 2/2 | 74.3 | 2/2 |
| Llama 3.0 8B | 75.8 | 81.5 | 2/2 | 79.4 | 2/2 |
| Llama 3.0 70B | 92.5 | 89.3 | 2/2 | 77.6 | 2/2 |
| Llama 3.1 8B | 76.6 | 82.5 | 2/2 | 77.6 | 2/2 |
| Llama 3.1 70B | 93.2 | 90.7 | 2/2 | 79.3 | 2/2 |
| Mistral 7B | 70.7 | 72.1 | 2/2 | 68.9 | 2/2 |
| Mixtral 8x7B | 83.4 | 78.8 | 2/2 | 65.9 | 2/2 |
| SOLAR 10.7B | 79.5 | 76.8 | 2/2 | 71.1 | 2/2 |
| Yi 6B | 65.8 | 75.1 | 2/2 | 60.8 | 2/2 |
| Yi 34B | 82.4 | 84.9 | 2/2 | 74.2 | 2/2 |
| GPT-3.5 Turbo | 83.6 | 84.5 | 2/2 | | |
| GPT-4o | 96.3 | 91.9 | 2/2 | | |

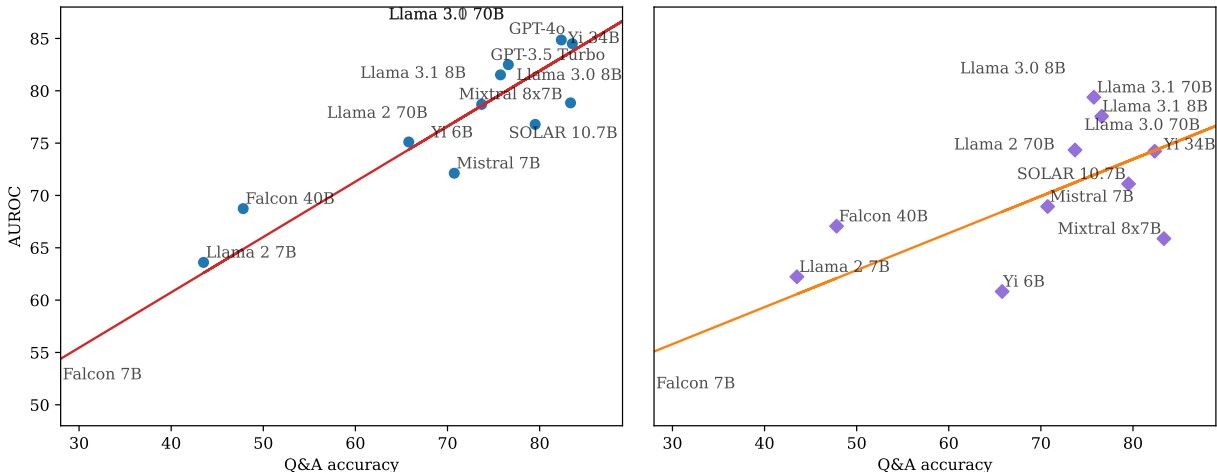

Figure 20: AUROC results for ARC-Challenge for MSP (left) and Max Logit (right). Both MSP AUROC and Max Logit AUROC exhibit significant correlations with Q&A accuracy: $R^2 = 0.93$ ($p < 10^{-4}$) and $R^2 = 0.68$ ($p < 10^{-3}$), respectively.

Table 18: Q&A-with-abstention results for ARC-Challenge. See Table 4 for an explanation of the scoring scheme.

| | Balanced | | | Conservative | | |
| LLM | No abstain | MSP | Max Logit | No abstain | MSP | Max Logit |
| --- | --- | --- | --- | --- | --- | --- |
| Falcon 7B | −45.7 | −3.6 | −1.6 | −118.5 | −9.8 | −4.3 |
| Falcon 40B | −4.3 | 10.3 | 6.2 | −56.4 | 6.0 | 0.3 |
| Llama 2 7B | −12.9 | 2.9 | 3.7 | −69.4 | −0.7 | 1.3 |
| Llama 2 70B | 47.4 | 48.4 | 49.6 | 21.2 | 39.7 | 34.0 |
| Llama 3.0 8B | 51.5 | 53.7 | 52.1 | 27.2 | 43.9 | 39.8 |
| Llama 3.0 70B | 85.0 | 85.0 | 85.0 | 77.5 | 77.5 | 77.5 |
| Llama 3.1 8B | 53.2 | 54.2 | 52.9 | 29.8 | 40.4 | 39.5 |
| Llama 3.1 70B | 86.4 | 86.7 | 86.4 | 79.7 | 81.5 | 79.7 |
| Mistral 7B | 41.6 | 44.4 | 37.5 | 12.3 | 25.9 | 23.7 |
| Mixtral 8x7B | 66.9 | 67.8 | 41.7 | 50.3 | 53.6 | 36.1 |
| SOLAR 10.7B | 59.0 | 59.9 | 59.0 | 38.5 | 43.6 | 43.0 |
| Yi 6B | 31.6 | 34.2 | 31.6 | −2.7 | 24.2 | 9.3 |
| Yi 34B | 64.7 | 65.4 | 59.9 | 47.1 | 49.4 | 50.1 |
| GPT-3.5 Turbo | 67.1 | 67.7 | | 50.7 | 58.4 | |
| GPT-4o | 92.6 | 92.6 | | 88.8 | 88.8 | |

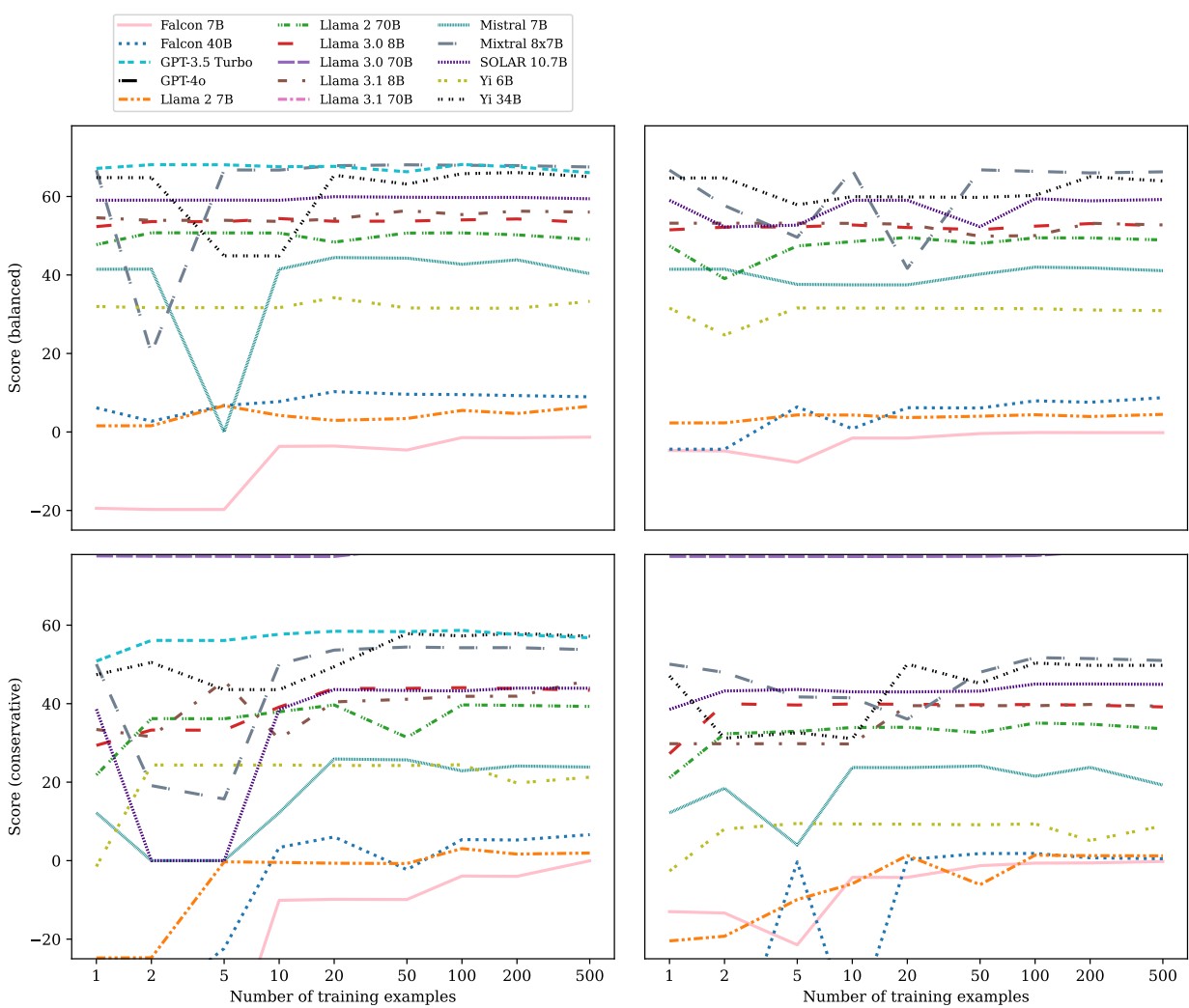

Figure 21: Q&A-with-abstention scores for ARC-Challenge as a function of the amount of training data. The x-axis is the number of data points included in the training data (referred to as $k$ in Section 6.1).

## E.2 HellaSwag

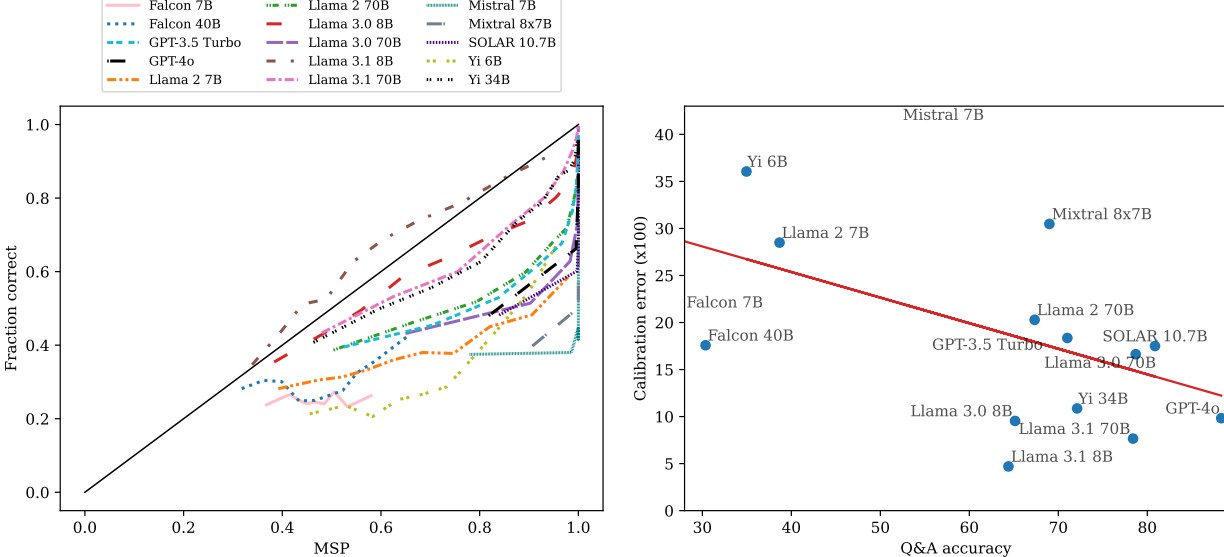

Figure 22: Calibration results for HellaSwag. The coefficient of determination between Q&A accuracy and calibration error was $R^2 = 0.23$ ($p = 0.07$).

Table 19: AUROC results for HellaSwag. See Table 2 for an explanation of the $p$-values.

| LLM | Q&A accuracy | MSP AUROC | MSP $p < 10^{-4}$ | Max Logit AUROC | Max Logit $p < 10^{-4}$ |
|---|---|---|---|---|---|
| Falcon 7B | 25.0 | 50.3 | 0/2 | 49.9 | 0/2 |
| Falcon 40B | 30.4 | 55.0 | 1/2 | 60.5 | 2/2 |
| Llama 2 7B | 38.7 | 60.3 | 2/2 | 57.8 | 2/2 |
| Llama 2 70B | 67.3 | 71.8 | 2/2 | 69.7 | 2/2 |
| Llama 3.0 8B | 65.1 | 72.4 | 2/2 | 69.8 | 2/2 |
| Llama 3.0 70B | 78.7 | 82.3 | 2/2 | 68.3 | 2/2 |
| Llama 3.1 8B | 64.4 | 72.4 | 2/2 | 67.2 | 2/2 |
| Llama 3.1 70B | 78.4 | 82.1 | 2/2 | 61.9 | 2/2 |
| Mistral 7B | 52.5 | 62.7 | 2/2 | 62.4 | 2/2 |
| Mixtral 8x7B | 69.0 | 68.3 | 2/2 | 58.3 | 2/2 |
| SOLAR 10.7B | 80.9 | 76.4 | 2/2 | 72.1 | 2/2 |
| Yi 6B | 35.0 | 68.0 | 2/2 | 62.7 | 2/2 |
| Yi 34B | 72.1 | 75.5 | 2/2 | 70.3 | 2/2 |
| GPT-3.5 Turbo | 71.0 | 77.8 | 2/2 | | |
| GPT-4o | 88.3 | 89.7 | 2/2 | | |

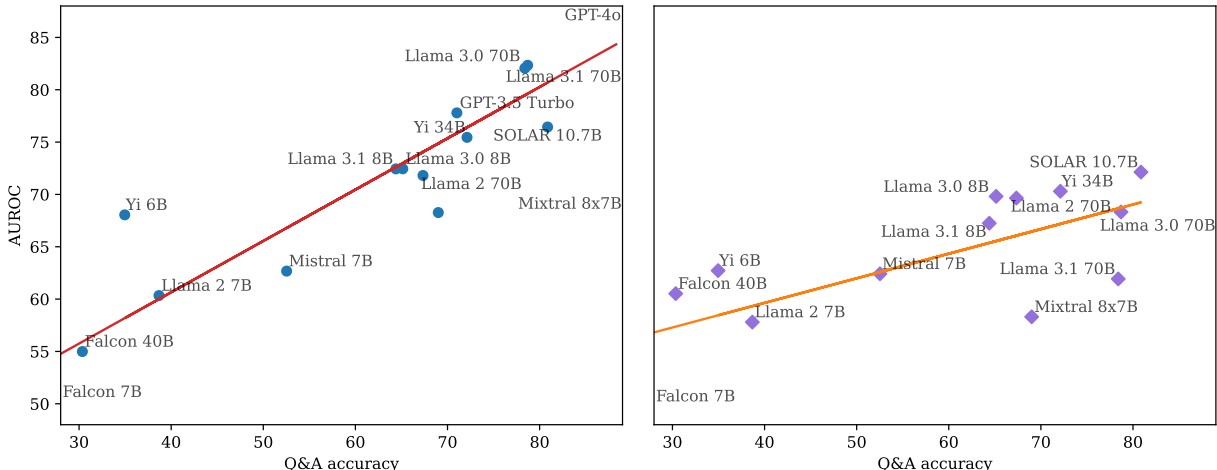

Figure 23: AUROC results for HellaSwag for MSP (left) and Max Logit (right). Both MSP AUROC and Max Logit AUROC exhibit significant correlations with Q&A accuracy: $R^2 = 0.85$ ($p < 10^{-4}$) and $R^2 = 0.52$ ($p = 0.005$), respectively.

Table 20: Q&A-with-abstention results for HellaSwag. See Table 4 for an explanation of the scoring scheme.

| | Balanced | | | Conservative | | |
|---|---|---|---|---|---|---|
| LLM | No abstain | MSP | Max Logit | No abstain | MSP | Max Logit |
| Falcon 7B | $-49.9$ | $-1.3$ | $-14.4$ | $-124.9$ | $-3.6$ | $-36.2$ |
| Falcon 40B | $-39.2$ | $-1.0$ | $0.1$ | $-108.9$ | $-5.6$ | $-4.9$ |
| Llama 2 7B | $-22.6$ | $1.5$ | $0.1$ | $-83.9$ | $-1.8$ | $0.0$ |
| Llama 2 70B | $34.7$ | $35.9$ | $34.9$ | $2.0$ | $21.8$ | $16.7$ |
| Llama 3.0 8B | $30.3$ | $34.4$ | $30.3$ | $-4.6$ | $18.1$ | $13.6$ |
| Llama 3.0 70B | $57.4$ | $57.4$ | $56.2$ | $36.1$ | $43.2$ | $38.6$ |
| Llama 3.1 8B | $28.8$ | $32.1$ | $29.5$ | $-6.8$ | $10.5$ | $10.1$ |
| Llama 3.1 70B | $56.8$ | $57.6$ | $56.8$ | $35.2$ | $38.8$ | $33.1$ |
| Mistral 7B | $5.1$ | $6.6$ | $10.0$ | $-42.4$ | $0.0$ | $-4.9$ |
| Mixtral 8x7B | $38.1$ | $39.7$ | $12.9$ | $7.1$ | $5.6$ | $6.0$ |
| SOLAR 10.7B | $61.7$ | $61.7$ | $61.7$ | $42.6$ | $42.6$ | $42.6$ |
| Yi 6B | $-30.0$ | $2.9$ | $0.4$ | $-95.1$ | $1.2$ | $0.0$ |
| Yi 34B | $44.3$ | $45.8$ | $37.0$ | $16.4$ | $31.1$ | $23.8$ |
| GPT-3.5 Turbo | $42.0$ | $43.6$ | | $13.1$ | $32.3$ | |
| GPT-4o | $76.6$ | $76.6$ | | $64.9$ | $64.9$ | |

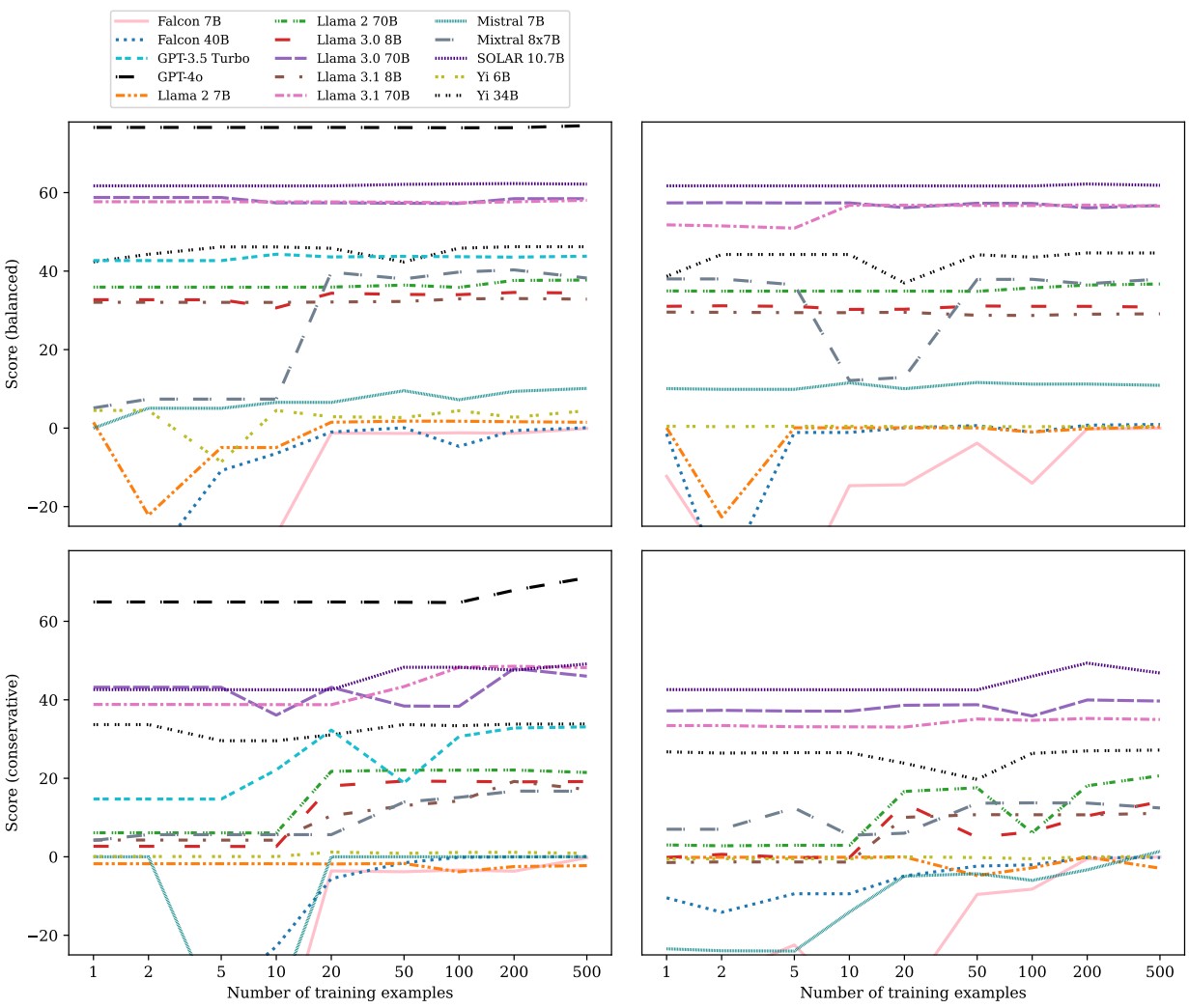

Figure 24: Q&A-with-abstention scores for HellaSwag as a function of the amount of training data. The x-axis is the number of data points included in the training data (referred to as $k$ in Section 6.1).

### E.3 MMLU

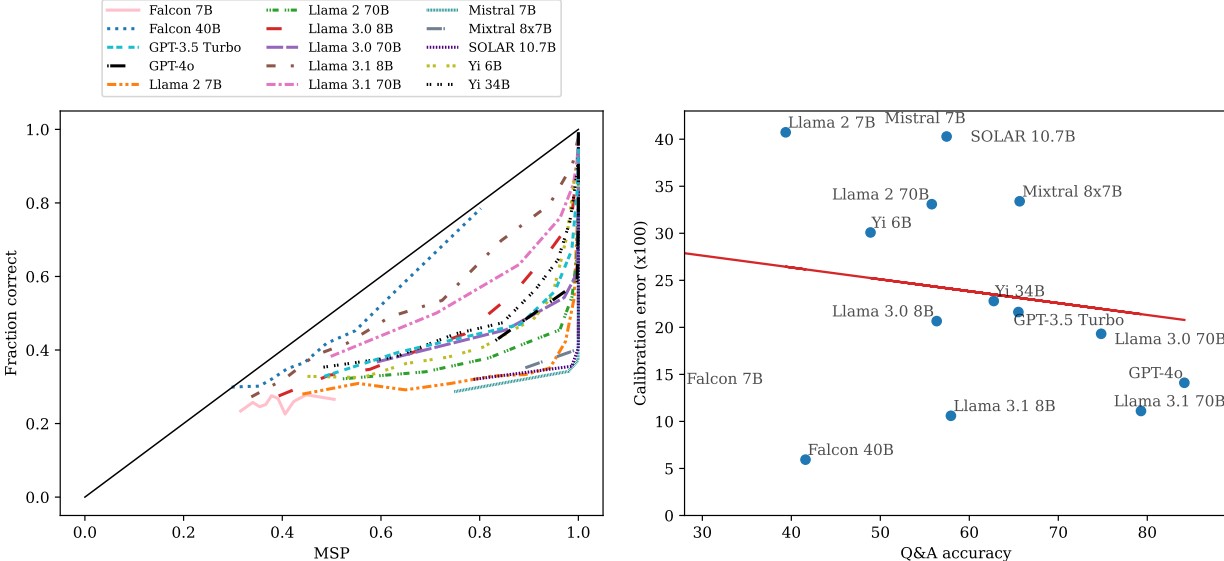

Figure 25: Calibration results for MMLU. The coefficient of determination between Q&A accuracy and calibration error was $R^2 = 0.03$ ($p = 0.57$).

Table 21: AUROC results for MMLU. See Table 2 for an explanation of the $p$-values.

| LLM | Q&A accuracy | MSP AUROC | MSP $p < 10^{-4}$ | Max Logit AUROC | Max Logit $p < 10^{-4}$ |
|---|---|---|---|---|---|
| Falcon 7B | 25.6 | 51.1 | 0/2 | 50.7 | 0/2 |
| Falcon 40B | 41.6 | 66.0 | 2/2 | 60.3 | 2/2 |
| Llama 2 7B | 39.4 | 64.5 | 2/2 | 61.7 | 2/2 |
| Llama 2 70B | 55.8 | 72.7 | 2/2 | 69.8 | 2/2 |
| Llama 3.0 8B | 56.3 | 77.0 | 2/2 | 76.1 | 2/2 |
| Llama 3.0 70B | 74.8 | 83.0 | 2/2 | 74.6 | 2/2 |
| Llama 3.1 8B | 57.9 | 77.7 | 2/2 | 73.4 | 2/2 |
| Llama 3.1 70B | 79.3 | 84.3 | 2/2 | 75.1 | 2/2 |
| Mistral 7B | 53.1 | 68.9 | 2/2 | 64.8 | 2/2 |
| Mixtral 8x7B | 65.7 | 74.8 | 2/2 | 65.3 | 2/2 |
| SOLAR 10.7B | 57.4 | 70.9 | 2/2 | 65.7 | 2/2 |
| Yi 6B | 48.9 | 68.3 | 2/2 | 61.8 | 2/2 |
| Yi 34B | 62.8 | 75.2 | 2/2 | 69.7 | 2/2 |
| GPT-3.5 Turbo | 65.5 | 80.1 | 2/2 | | |
| GPT-4o | 84.2 | 85.6 | 2/2 | | |

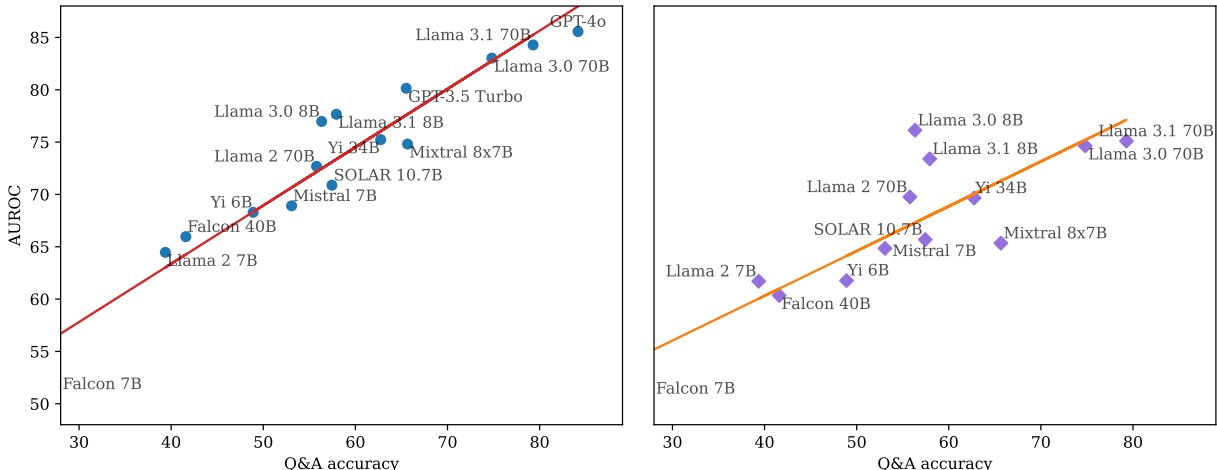

Figure 26: AUROC results for MMLU for MSP (left) and Max Logit (right). Both MSP AUROC and Max Logit AUROC exhibit significant correlations with Q&A accuracy: $R^2 = 0.92$ ($p < 10^{-4}$) and $R^2 = 0.72$ ($p < 10^{-3}$), respectively.

Table 22: Q&A-with-abstention results for MMLU. See Table 4 for an explanation of the scoring scheme.

| LLM | Balanced | | | Conservative | | |
| --- | --- | --- | --- | --- | --- | --- |
| | No abstain | MSP | Max Logit | No abstain | MSP | Max Logit |
| Falcon 7B | −48.7 | −3.2 | −2.7 | −123.1 | −1.0 | −7.0 |
| Falcon 40B | −16.9 | −1.2 | −11.0 | −75.3 | 3.0 | 0.2 |
| Llama 2 7B | −21.3 | 6.2 | 2.8 | −81.9 | 1.3 | −3.5 |
| Llama 2 70B | 11.5 | 11.5 | 11.5 | −32.7 | 13.8 | −32.7 |
| Llama 3.0 8B | 12.6 | 14.2 | 14.7 | −31.0 | 18.1 | 14.2 |
| Llama 3.0 70B | 49.6 | 49.6 | 49.6 | 24.4 | 43.7 | 24.4 |
| Llama 3.1 8B | 15.9 | 27.9 | 18.5 | −26.2 | 19.9 | −17.9 |
| Llama 3.1 70B | 58.6 | 61.2 | 58.6 | 37.9 | 47.5 | 43.1 |
| Mistral 7B | 6.1 | 6.1 | 9.4 | −40.9 | −20.8 | −30.1 |
| Mixtral 8x7B | 31.2 | 31.2 | 31.4 | −3.1 | −3.1 | −2.6 |
| SOLAR 10.7B | 14.9 | 21.5 | 20.6 | −27.6 | 0.0 | 5.5 |
| Yi 6B | −2.2 | 11.2 | 0.9 | −53.4 | 3.6 | −3.7 |
| Yi 34B | 25.5 | 31.1 | 29.3 | −11.8 | 7.9 | 14.5 |
| GPT-3.5 Turbo | 31.0 | 38.1 | | −3.5 | 30.1 | |
| GPT-4o | 68.3 | 68.5 | | 52.5 | 53.0 | |

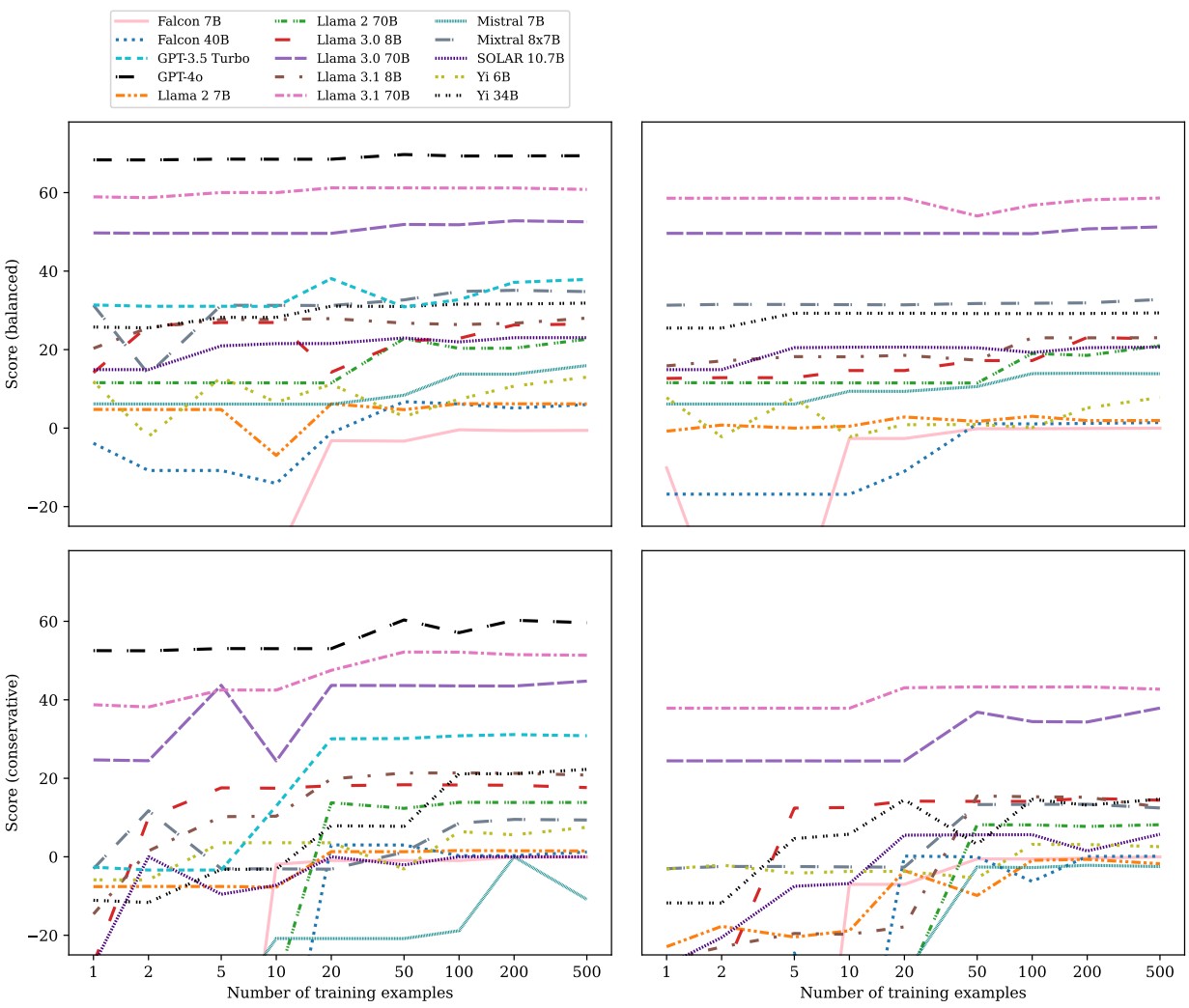

Figure 27: Q&A-with-abstention scores for MMLU as a function of the amount of training data. The x-axis is the number of data points included in the training data (referred to as $k$ in Section 6.1).

### E.4  TruthfulQA

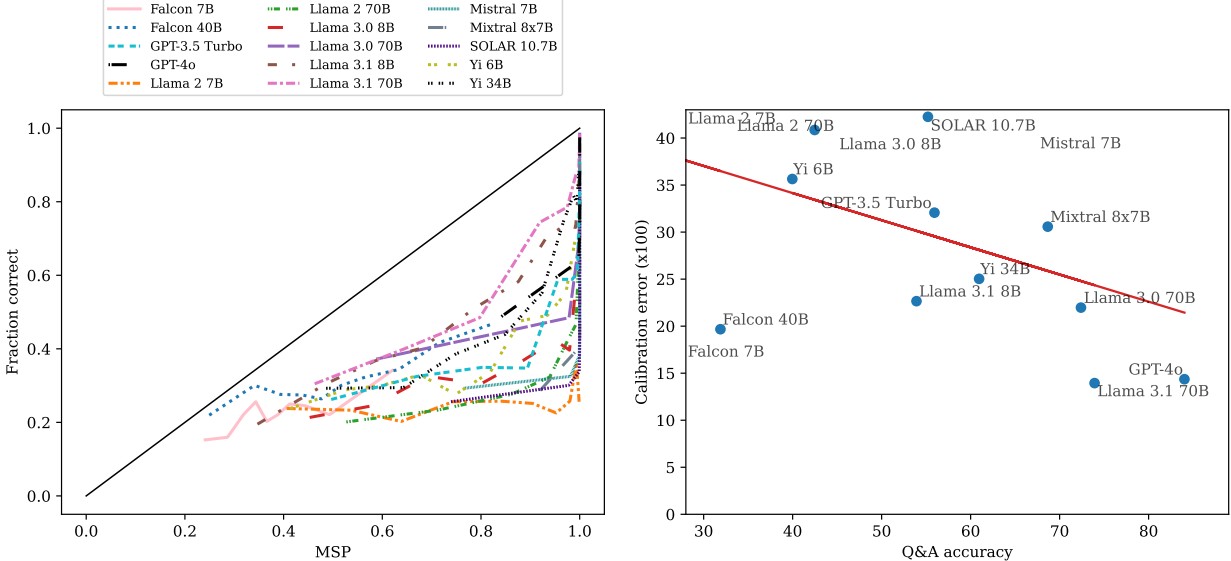

Figure 28: Calibration results for TruthfulQA. The coefficient of determination between Q&A accuracy and calibration error was $R^2 = 0.17$ ($p = 0.13$).

Table 23: AUROC results for TruthfulQA. See Table 2 for an explanation of the $p$-values.

| | | MSP | | Max Logit | |
| LLM | Q&A accuracy | AUROC | $p < 10^{-4}$ | AUROC | $p < 10^{-4}$ |
| --- | --- | --- | --- | --- | --- |
| Falcon 7B | 22.7 | 56.5 | 1/2 | 57.2 | 1/2 |
| Falcon 40B | 31.9 | 59.3 | 2/2 | 55.0 | 1/2 |
| Llama 2 7B | 25.4 | 53.3 | 0/2 | 57.2 | 1/2 |
| Llama 2 70B | 46.3 | 72.2 | 2/2 | 67.5 | 2/2 |
| Llama 3.0 8B | 42.5 | 69.1 | 2/2 | 62.9 | 2/2 |
| Llama 3.0 70B | 72.4 | 78.6 | 2/2 | 66.8 | 2/2 |
| Llama 3.1 8B | 53.9 | 74.0 | 2/2 | 62.2 | 2/2 |
| Llama 3.1 70B | 73.9 | 84.1 | 2/2 | 67.7 | 2/2 |
| Mistral 7B | 55.2 | 71.1 | 2/2 | 63.4 | 2/2 |
| Mixtral 8x7B | 68.7 | 73.8 | 2/2 | 62.7 | 2/2 |
| SOLAR 10.7B | 52.4 | 72.0 | 2/2 | 70.0 | 2/2 |
| Yi 6B | 40.0 | 65.8 | 2/2 | 61.2 | 2/2 |
| Yi 34B | 61.0 | 77.5 | 2/2 | 68.5 | 2/2 |
| GPT-3.5 Turbo | 55.9 | 75.3 | 2/2 | | |
| GPT-4o | 84.0 | 82.9 | 2/2 | | |

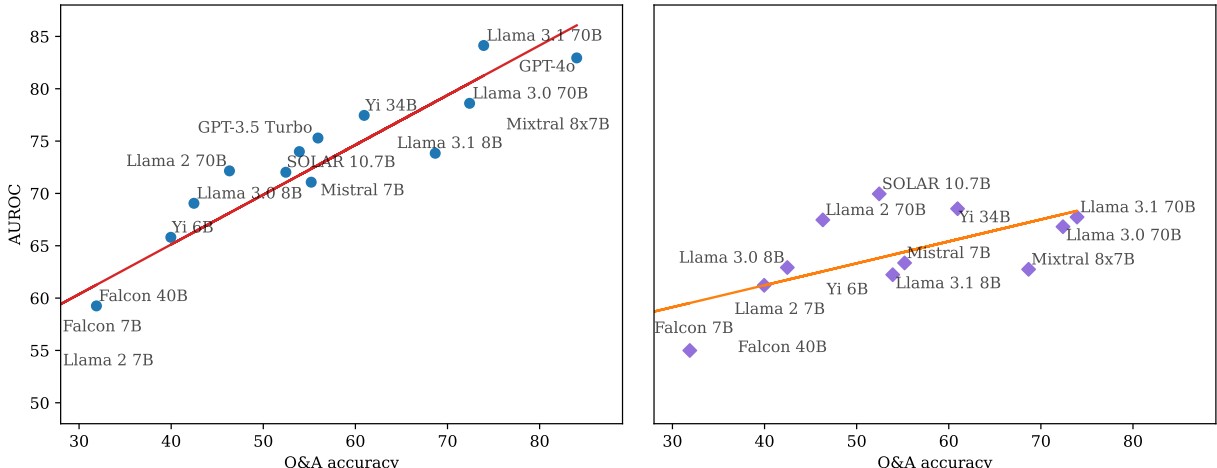

Figure 29: AUROC results for TruthfulQA for MSP (left) and Max Logit (right). Both MSP AUROC and Max Logit AUROC exhibit significant correlations with Q&A accuracy: $R^2 = 0.90$ ($p < 10^{-4}$) and $R^2 = 0.56$ ($p = 0.004$), respectively.

Table 24: Q&A-with-abstention results for TruthfulQA. See Table 4 for an explanation of the scoring scheme.

| | Balanced | | | Conservative | | |
|---|---|---|---|---|---|---|
| LLM | No abstain | MSP | Max Logit | No abstain | MSP | Max Logit |
| Falcon 7B | −54.8 | −17.0 | −20.9 | −132.2 | −23.7 | −14.4 |
| Falcon 40B | −36.2 | −0.7 | −4.5 | −104.3 | −1.0 | −11.9 |
| Llama 2 7B | −49.6 | −29.1 | −3.0 | −124.3 | −6.3 | −6.1 |
| Llama 2 70B | −7.4 | 14.1 | 4.0 | −61.2 | −0.5 | −0.6 |
| Llama 3.0 8B | −15.2 | −2.5 | 1.7 | −72.9 | 1.4 | −8.2 |
| Llama 3.0 70B | 44.9 | 46.1 | 44.9 | 17.3 | 37.4 | 18.8 |
| Llama 3.1 8B | 7.7 | 21.9 | 9.6 | −38.5 | 5.8 | −2.2 |
| Llama 3.1 70B | 48.0 | 49.4 | 48.0 | 21.9 | 38.9 | 26.6 |
| Mistral 7B | 10.2 | 18.9 | 11.0 | −34.8 | −9.9 | 2.0 |
| Mixtral 8x7B | 37.3 | 40.0 | 37.3 | 5.9 | 13.1 | 14.9 |
| SOLAR 10.7B | 4.7 | 8.4 | 17.7 | −42.9 | −33.9 | 7.1 |
| Yi 6B | −20.4 | −6.6 | −17.8 | −80.7 | 1.4 | −26.7 |
| Yi 34B | 21.7 | 22.1 | 26.7 | −17.5 | 24.0 | 4.8 |
| GPT-3.5 Turbo | 11.8 | 24.5 | | −32.3 | 11.6 | |
| GPT-4o | 68.2 | 65.3 | | 52.2 | 58.1 | |

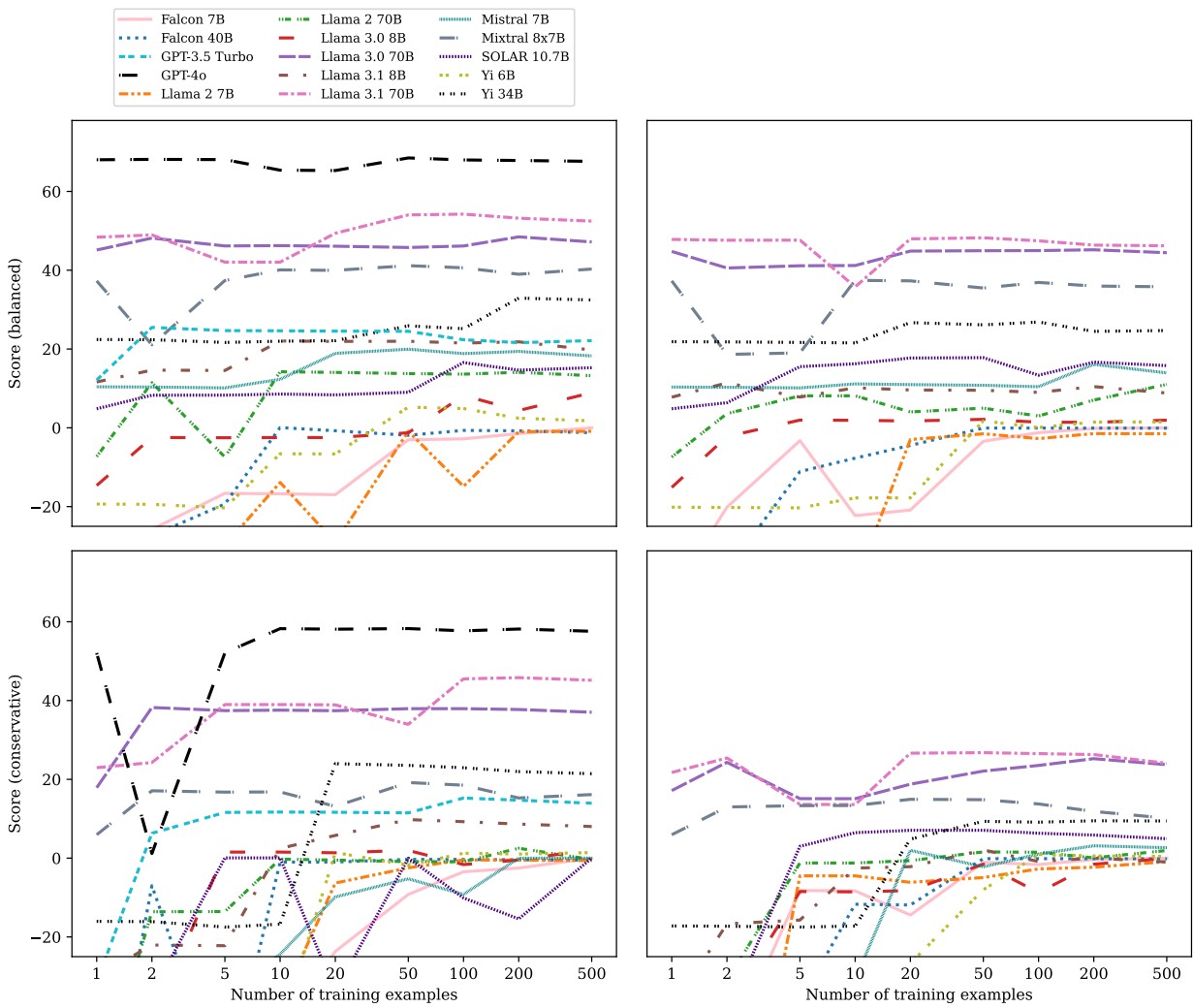

Figure 30: Q&A-with-abstention scores for TruthfulQA as a function of the amount of training data. The x-axis is the number of data points included in the training data (referred to as $k$ in Section 6.1).

## E.5 WinoGrande

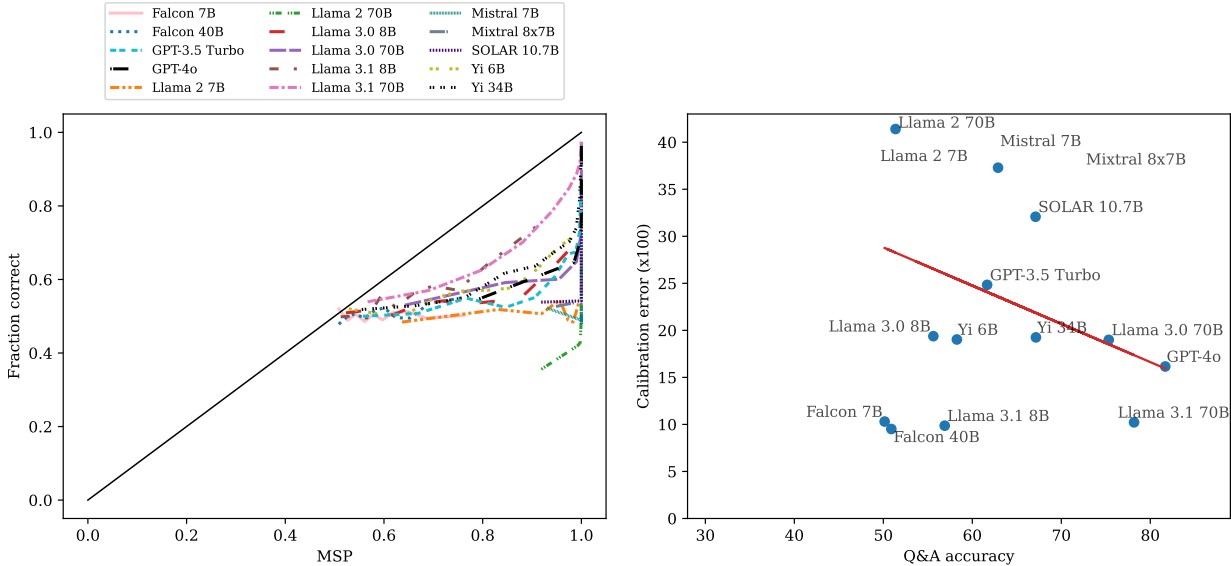

Figure 31: Calibration results for WinoGrande. The coefficient of determination between Q&A accuracy and calibration error was $R^2 = 0.10$ ($p = 0.25$).

Table 25: AUROC results for WinoGrande. See Table 2 for an explanation of the $p$-values.

| LLM | Q&A accuracy | MSP | | Max Logit | |
|---|---|---|---|---|---|
| | | AUROC | $p < 10^{-4}$ | AUROC | $p < 10^{-4}$ |
| Falcon 7B | 50.2 | 49.9 | 0/2 | 50.6 | 0/2 |
| Falcon 40B | 50.9 | 52.0 | 0/2 | 50.1 | 0/2 |
| Llama 2 7B | 51.4 | 51.1 | 0/2 | 50.3 | 0/2 |
| Llama 2 70B | 50.4 | 60.1 | 2/2 | 53.2 | 1/2 |
| Llama 3.0 8B | 55.6 | 55.8 | 2/2 | 56.4 | 2/2 |
| Llama 3.0 70B | 75.3 | 73.3 | 2/2 | 54.6 | 1/2 |
| Llama 3.1 8B | 56.9 | 58.1 | 2/2 | 55.8 | 2/2 |
| Llama 3.1 70B | 78.2 | 75.9 | 2/2 | 63.9 | 2/2 |
| Mistral 7B | 53.6 | 55.9 | 2/2 | 54.3 | 2/2 |
| Mixtral 8x7B | 62.9 | 54.2 | 2/2 | 50.1 | 0/2 |
| SOLAR 10.7B | 67.1 | 63.0 | 2/2 | 55.3 | 2/2 |
| Yi 6B | 58.3 | 57.8 | 2/2 | 55.7 | 2/2 |
| Yi 34B | 67.2 | 66.0 | 2/2 | 62.4 | 2/2 |
| GPT-3.5 Turbo | 61.7 | 62.8 | 2/2 | | |
| GPT-4o | 81.7 | 76.4 | 2/2 | | |

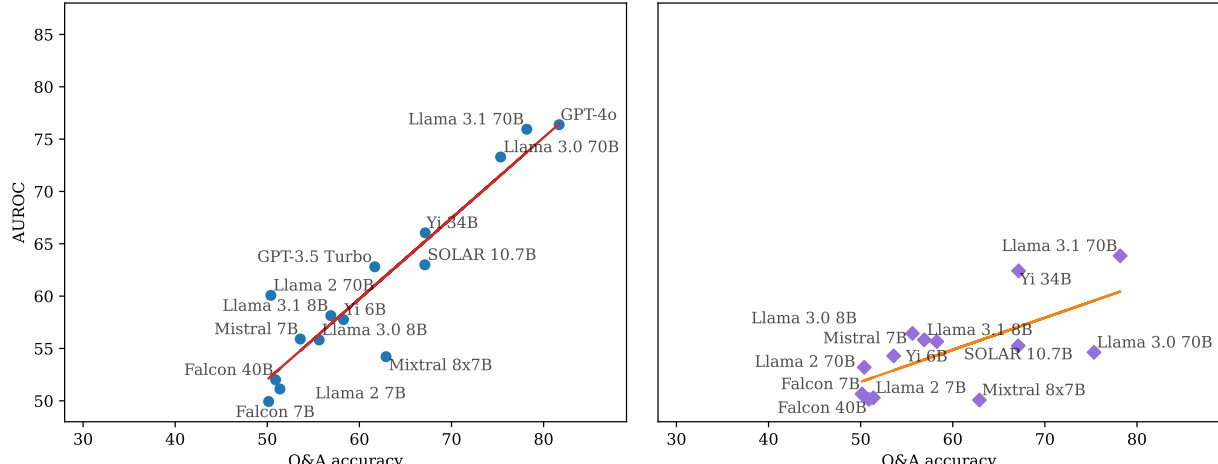

Figure 32: AUROC results for WinoGrande for MSP (left) and Max Logit (right). MSP AUROC exhibits a significant correlation with Q&A accuracy ($R^2 = 0.86, p < 10^{-4}$). However, the correlation between Max Logit AUROC and Q&A accuracy is much weaker ($R^2 = 0.45, p < 10^{-3}$).

Table 26: Q&A-with-abstention results for WinoGrande. See Table 4 for an explanation of the scoring scheme.

| | Balanced | | | Conservative | | |
|---|---|---|---|---|---|---|
| LLM | No abstain | MSP | Max Logit | No abstain | MSP | Max Logit |
| Falcon 7B | 0.2 | 0.1 | 0.2 | $-49.7$ | $-39.6$ | $-29.7$ |
| Falcon 40B | 1.8 | 1.8 | 1.1 | $-47.3$ | $-1.7$ | $-19.8$ |
| Llama 2 7B | 2.7 | 2.7 | 2.7 | $-45.9$ | $-8.8$ | $-12.7$ |
| Llama 2 70B | 0.8 | 7.4 | 0.8 | $-48.9$ | $-5.5$ | $-0.9$ |
| Llama 3.0 8B | 11.3 | 11.1 | 6.3 | $-33.1$ | 1.1 | $-0.1$ |
| Llama 3.0 70B | 50.7 | 49.7 | 47.5 | 26.0 | 31.0 | 21.7 |
| Llama 3.1 8B | 13.8 | 14.0 | 8.6 | $-29.2$ | 1.8 | $-6.5$ |
| Llama 3.1 70B | 56.3 | 56.2 | 54.1 | 34.5 | 42.0 | 34.6 |
| Mistral 7B | 7.2 | 0.0 | 6.3 | $-39.2$ | 0.0 | 0.7 |
| Mixtral 8x7B | 25.8 | 25.8 | 25.0 | $-11.3$ | $-11.3$ | $-10.2$ |
| SOLAR 10.7B | 34.2 | 34.2 | 34.2 | 1.3 | 1.3 | 1.3 |
| Yi 6B | 16.5 | 15.4 | 10.6 | $-25.2$ | 1.6 | 0.5 |
| Yi 34B | 34.3 | 33.4 | 32.2 | 1.4 | 9.4 | 6.6 |
| GPT-3.5 Turbo | 23.4 | 23.4 | | $-15.0$ | 7.7 | |
| GPT-4o | 63.3 | 63.3 | | 45.0 | 45.0 | |

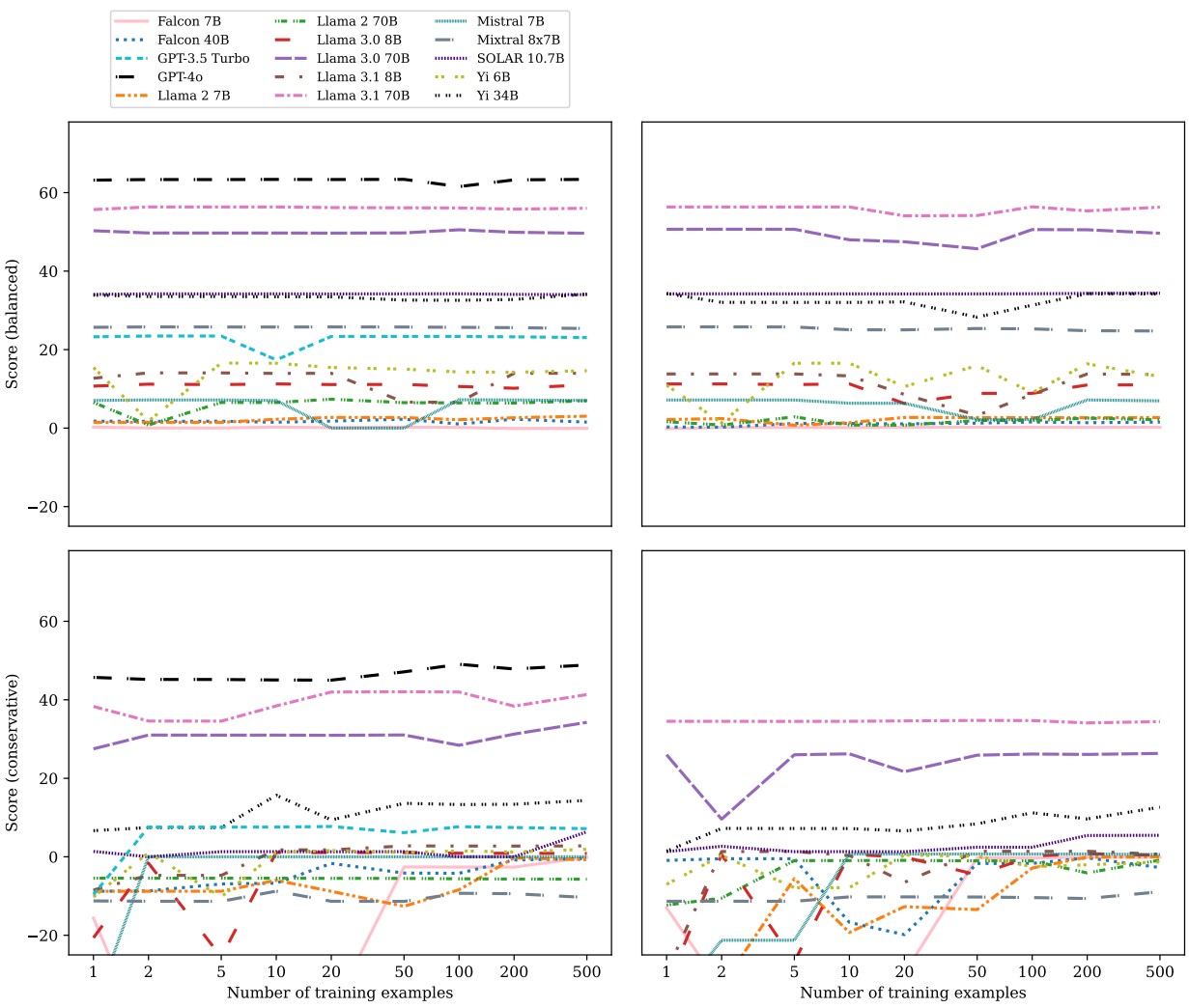

Figure 33: Q&A-with-abstention scores for WinoGrande as a function of the amount of training data. The x-axis is the number of data points included in the training data (referred to as $k$ in Section 6.1).

