# OpenReview forum: "Probabilities of Chat LLMs Are Miscalibrated but Still Predict Correctness on Multiple-Choice Q&A"
_TMLR — Accepted by TMLR_

### Review · Reviewer_PGqA · 2025-05-27

**Summary Of Contributions:**

The paper measures an LLMs uncertainty on multiple choice benchmarks by searching the LLMs response to the MCQ for A./B./etc and measuring softmax-normalized probability over the letters for that position. They call this the MSP. It finds that instruction-tuned models are overconfident, and this overconfidence doesnt decrease with model capability. They then show that a model's MSP can be predictive of its correctness, and use this to make an abstention classifier.

**Audience:**

Yes

**Claims And Evidence:**

Yes

**Requested Changes:**

1. Include a description of how MSP is computed in the introduction, as it is non-standard.
2. Report experiments with the typical way (for eg as implemented in LM Eval Harness) of assigning probabilities to options in mcqs, based on normalized log-likelihoods over the full option string [1]. This also allows extending the study to base models as format-following will no longer be needed. It is well known that base models are more calibrated than instruct versions, and that base model calibation improves with capability/scale, so this is important [2, 3].
3. Compare to prior work on verbalized uncertainty in LLMs [4]

[1] Biderman, S., Schoelkopf, H., Sutawika, L., Gao, L., Tow, J., Abbasi, B., ... & Zou, A. (2024). Lessons from the trenches on reproducible evaluation of language models. arXiv preprint arXiv:2405.14782.

[2] He, G., Cui, P., Chen, J., Hu, W., & Zhu, J. (2023). Investigating uncertainty calibration of aligned language models under the multiple-choice setting. arXiv preprint arXiv:2310.11732.

[3] Achiam, J., Adler, S., Agarwal, S., Ahmad, L., Akkaya, I., Aleman, F. L., ... & McGrew, B. (2023). Gpt-4 technical report. arXiv preprint arXiv:2303.08774.

[4] Xiong, M., Hu, Z., Lu, X., Li, Y., Fu, J., He, J., & Hooi, B. (2023). Can llms express their uncertainty? an empirical evaluation of confidence elicitation in llms. ICLR 2024.

**Strengths And Weaknesses:**

Strengths:
1. A comprehensive replication of known results about LLM multiple choice calibation across instruction-tuned models.
2. Demonstrate how performance changes when the multiple choice probability is used to train a separate abstention classifier.

Weaknesses:
1. The key results have been shown before.
2. Non-standard way to compute choice probabilities

---

> ### Author Response · Authors · 2025-05-29
> **Two follow-up questions**
>
> We thank the reviewer for helpful feedback and believe that all of the recommended changes will strengthen the paper. We will make Requested Changes 1 and 3 as stated. We will also make Requested Change 2 to the extent possible, although there is an obstacle.
>
> ## Requested Change 2
>
> This is the most important change in our opinion, as it is a key methodological change that will better align our approach with the literature. The recommendation in Appendix A.2 of [1] is to do the following: construct the entire answer string for each of the answer choices (e.g., “A. lightning”, “B. cellular respiration”, etc), compute the (conditional) probability for each of these strings, and choose the one with the highest probability. Unfortunately, this is no longer possible in the OpenAI API (it was previously possible with the legacy Completions API, which does not support modern models). Now, one can only compute the probability of the generated string, not an arbitrary prompt string.
>
> This leaves us with two main options:
> 1. Rather than computing the probability for the whole answer string (e.g., “A. lightning”), compute the probability that “A”/“B”/“C”/etc is the first token. Then choose the answer token with the highest probability.
>
> 2. Follow Appendix A.2 of [1] for the open-source models, but do Option 1 for the OpenAI models.
>
> We advocate for Option 1 so that all models in the study are evaluated with the same methodology. However, we are also open to Option 2 if the reviewer insists. Note that both approaches potentially allow the inclusion of base models in the study since format-following will not be needed (although there could still be concerns if, e.g., a base model assigns very low probability to all of the answers).
>
> **Follow-up question 1: Is the reviewer okay with Option 1 above?**
>
> ## Which results have been previously shown
>
> The reviewer states that the key results have been shown before. We agree that it has previously been shown that fine-tuned LLMs are miscalibrated (and in particular, overconfident). However, we are not aware of prior work that shows either of these two results, which are our main results:
>
> 1. Calibration does *not* improve with model capability
>
> 2. The ability of the MSP to predict correctness *does* improve with model capability
>
> **Follow-up question 2: Is the reviewer aware of prior work which shows these results?** If so, we would appreciate references so that we can cite them in our paper. Either way, when revising we will make it clearer which of our results are novel.

---

> > ### Comment · Reviewer_PGqA · 2025-05-31
> > **Thanks for the questions**
> >
> > I am fine with option 1 to ensure uniform evaluations across models.
> >
> > I find it unlikely that claim 1 (calibration does not improve with capability) holds for base models. Please run the same experiments on base models and see if the trend still holds, or otherwise qualify the claims as being a property of existing instruction tuning methodologies.
> >
> > Does the second claim essentially mean that the model tends to have higher confidence for questions it gets right, even if the absolute value is not precisely calibrated? If so, I think this is evident from the calibration curves in many papers.

---

> > > ### Author Response · Authors · 2025-06-02
> > > **Clarifications**
> > >
> > > Thank you for the follow-up. We will proceed with Option 1 for the methodology.
> > >
> > > **Claim 1:** To clarify, all of the results in the submitted version of the paper are for fine-tuned models only, including our finding that calibration does not improve with model capability. (Here “capability” specifically means model accuracy on the underlying Q&A task.) We will make this clearer when revising.
> > >
> > > We agree that this claim is unlikely to hold for base models, and this is something we should be able to confirm with the new set of experiments that will include base models. We appreciate this suggestion.
> > >
> > > **Claim 2:** We believe there is a misunderstanding here. Consider the following two questions:
> > >
> > >  A. For a particular model, does the model tend to have higher confidence for questions it gets right (even if the absolute value is not precisely calibrated)? The strength of this correlation is measured by AUROC, and we compute one AUROC value per model (just considering MSP and ignoring Max Logit). These AUROC values are shown in Table 1.
> > >
> > > B. How does the strength of this correlation (AUROC) vary across models as a function of model capability? This is shown in Figure 3.
> > >
> > > The reviewer’s comment refers to Question A. We agree that calibration plots in prior work suggest a positive answer to Question A (which our rigorous evaluation confirms). However, we do not believe Question B has been studied in prior work.
> > >
> > > Our experiments show that the AUROC value improves as model capability improves (Figure 3). In other words, not only are models pretty good at predicting correctness (Question A), but we can expect models to become even better at predicting correctness as model capabilities improve (Question B). In our opinion, this is our most significant finding because of its implications for future models that do not exist yet. This trend predicts that increasingly powerful future models would also increasingly improve at detecting when they are incorrect, which could be a useful tool in averting harmful responses.
> > >
> > > We will make the distinction between Questions A and B clearer when revising. We would also appreciate any feedback from the reviewer on how we can communicate this more clearly.

---

> > > > ### Comment · Reviewer_PGqA · 2025-06-04
> > > >
> > > > Thanks for the clarification. I see the point now. It's a subtle point which can be hard to internalize from the text. Having a Figure 1 that illustrates the summary of the findings (as currently listed in the Introduction) might help.
> > > >
> > > > Is an implication that post-hoc calibration techniques like platt scaling will lead to higher calibration also for instruct-tuned models? If this is true, I think it would be nice to test this, and mention the result as one of the implications of MSP being better at predicting correctness (if true).

---

> > > > > ### Author Response · Authors · 2025-06-05
> > > > >
> > > > > We really appreciate the reviewer's engagement with respect to questions A and B: this discussion helped us understand why this distinction may not be apparent to readers given the current text. The suggestion of a "summary of findings" figure makes sense, and we will incorporate that.
> > > > >
> > > > > The question regarding post-hoc calibration techniques is quite interesting, and is not something we previously considered. We think our results suggest (but not definitively imply) that the reviewer's conjecture holds. We will add an experiment where Platt scaling is applied to the MSPs and a calibration analysis is performed on the resulting values. We appreciate the suggestion.

---

> > > > > > ### Comment · Reviewer_zxok · 2025-06-08
> > > > > >
> > > > > > I have now read this thread.
> > > > > >
> > > > > > My current understanding is that:
> > > > > > 1. For pre-trained models, the calibration improves with size (e.g. Fig 4 right in Kadavath 2022 https://arxiv.org/abs/2207.05221).
> > > > > > 2. RLHF breaks the calibration (e.g. Fig 8 in GPT4 tech report https://arxiv.org/abs/2303.08774, Fig 1 in https://arxiv.org/abs/2505.01997).
> > > > > > 3. You can at least partially recover the calibration via temperature/Platt/other scaling method (e.g. Table 1 in https://arxiv.org/pdf/2403.08819, Table 1 in https://arxiv.org/abs/2409.19817).
> > > > > > 4. AUROC (as evaluated in "selective prediction" setup) is only rank-sensitive (and magnitude insensitive), hence: if 1 holds, then your Claim 2 follows.
> > > > > >
> > > > > > Please correct me/let's discuss this if you think differently!

---

> > > > > > > ### Comment · Reviewer_zxok · 2025-06-08
> > > > > > >
> > > > > > > Should say "improves with model-size/capability" - I'm thinking of it a bit exchangeably here, which might be a bit of a simplification here. Generally speaking, I agree "model capability" is a better "x-axis" for studying these scaling laws.

---

> > > > > > > > ### Comment · Reviewer_zxok · 2025-06-08
> > > > > > > >
> > > > > > > > [sorry, comments cannot be edited]
> > > > > > > >
> > > > > > > > My statement 4 needs to be amended to say something along the lines of:
> > > > > > > > "if 1 and 3 hold, and for 3 we assume e.g. some fixed fraction of the ECE gap w.r.t. the pretrained-base-model is recovered irrespective of model-capabilities, then your Claim 2 follows."
> > > > > > > >
> > > > > > > > Although assuming "some fixed fraction of the ECE gap w.r.t. the pretrained-base-model is recovered irrespective of model-capabilities" seems relatively mild to me, it shouldn't be taken for granted, and hence I think empirically evaluating Claim 2 is a valuable contribution.
> > > > > > > >
> > > > > > > > Overall, reiterating from my review, for me, the largest obstacle for accepting this work is lack of a more thorough credit assignment to prior works that already studied the calibration of RLHFed models and proposed methods to (at least partially) alleviate it (#3 and https://arxiv.org/abs/2505.01997 although that last one is clearly a concurrent work).

---

> > > > > > > > > ### Author Response · Authors · 2025-06-10
> > > > > > > > > **Follow-up on Claim 2**
> > > > > > > > >
> > > > > > > > > We appreciate the thoughtful discussion above, which will help us better position our results in the revised version of the paper. We think that the four-step reasoning proposed by Reviewer zxok is a clever hypothesis for why one might suspect Claim 2 to hold, but is far from conclusive. As the reviewer notes below, additional assumptions would be required for such a deduction, such as
> > > > > > > > > > some fixed fraction of the ECE gap w.r.t. the pretrained-base-model is recovered irrespective of model-capabilities
> > > > > > > > >
> > > > > > > > > We believe one would also need to assume the absence of other confounding effects. However, the reviewer’s reasoning does make sense intuitively. This reasoning also nicely connects our results to prior work, so we will include this when revising.

---

### Review · Reviewer_BWQa · 2025-05-30

**Summary Of Contributions:**

This work presents a thorough empirical evaluation of miscalibration in chat-based large language models (LLMs), and explores how useful uncertainty estimates can still be recovered despite such miscalibration. The authors assess 15 LLMs on a multiple-choice question answering (MCQ) task, finding consistent evidence of overconfidence in the maximum softmax probabilities (MSPs) produced by the models. They hypothesize that the magnitude of the MSP can serve as a predictor of answer correctness and provide strong empirical support for this hypothesis. Building on these findings, the authors demonstrate that an abstention mechanism—based on thresholding either the MSP or the maximum logit—can improve a weighted accuracy metric that penalizes incorrect predictions.

**Audience:**

Yes

**Claims And Evidence:**

Yes

**Requested Changes:**

Typo/formatting
- Final sentence of 2nd paragraph of “Models” in Section 3 contains a duplicate “that”. I would recommend deleting this sentence since it is speculative.
- Final sentence of “Minimal correlation between model size and AUROC” in section 5.2 ends abruptly, I believe this may be missing a word at the end.
- Fig 2 Left, Fig 7, Fig 8, Fig 9, Fig 11 legends cover some of the plot contents. I request moving the legend to outside of the plot.

Non-critical
- Seeing that k = 20 worked well for determining the threshold in the abstention experiment is quite surprising at first glance. Does the optimal value for k (or point of diminishing returns) vary much between datasets or is this more LLM dependent? I would be interested in such results and believe that they would strengthen the evidence for application of this method.

Critical: None

**Strengths And Weaknesses:**

Strengths

- The paper is well written and has a natural, logical flow. The related work section clearly situates the contribution within existing literature, and I particularly appreciated the thoughtful comparison to Kadavath et al. (2022). Key results are presented in the main text, with additional analyses appropriately deferred to the appendix.
- The experimental setup is thorough and well described. The authors evaluate 15 LLMs across 5 datasets and employ rigorous statistical testing, including a conservative multiple testing correction, which lends additional credibility to their findings.
- The questions addressed in this paper are relatively simple but highly relevant. The authors conduct a careful and thorough empirical evaluation to answer them convincingly. This kind of work—systematically investigating foundational questions through well-designed experiments—is valuable for the field. The paper demonstrates that while miscalibration is common in chat LLMs, the MSP still carries useful signal for uncertainty quantification.

Weaknesses

- One limitation of the analysis is the focus on the magnitude of the MSP as the sole predictor of response correctness,. While this is an intuitive score, it only provides a simple summary of model uncertainty and fails to capture potentially informative aspects of the full softmax prob distribution. In MCQs, the relative probabilities assigned to the other answer options can provide additional signal about model uncertainty or confusion. For instance, a distribution like (0.52, 0.16, 0.16, 0.16) reflects higher confidence in the top choice compared to a distribution like (0.52, 0.51, 0.01, 0.00), where the model appears uncertain between two options. Incorporating features that summarize the shape or entropy of the full distribution — or explicitly modeling the gap between the top two softmax scores — could improve calibration analysis and offer a more nuanced understanding of model confidence.

---

> ### Author Response · Authors · 2025-06-05
> **Response to Reviewer BWQa**
>
> We thank the reviewer for helpful feedback. We address the weakness and requested changes below.
>
> 1. We agree that it would be valuable to extend the experiments to other confidence measures like entropy and margin (the difference between the top two softmax scores). Although this was not specifically listed as a requested change, we will add entropy and margin to the correctness prediction (AUROC) analysis when revising (i.e., repeat all of the experiments described in Section 5 for entropy and margin).
>
>     We could also perform calibration analysis for these metrics, but the interpretation there is less clear. To us, calibration has a specific meaning: does the probability assigned to an outcome match the actual chance of that outcome? Entropy and margin do not correspond to the probabilities of any particular outcome, so calibration analysis may not be appropriate. Given this, we lean away from including calibration analysis for entropy and margin, but we can include this analysis if the reviewer disagrees with our reasoning.
>
> 2. We appreciate the typo/formatting pointers and will make these changes.
>
> 3. This is an interesting question. Based on our informal observations, the chosen thresholds in the abstention experiments vary significantly between models and much less significantly between datasets. However, we will look into this more and add some formal results to the paper (what are the specific thresholds for each model and what are the dataset-level averages). We will also include a plot visualizing the impact of k on the scores.
>
> Please let us know if you have any additional questions or if the proposed changes above do not fully address your concerns.

---

### Review · Reviewer_zxok · 2025-06-08

**Summary Of Contributions:**

The paper evaluates the calibration properties of chat/instruct-finetuned LLMs and conclude they are severely miscalibrated.
Further, they evaluate the Maximum Sequence Probability and Max-Logit scores in the selective-prediction framework and report the AUROC scores.
Lastly, they evaluate how many labelled samples are required to decide about the threshold value to achieve a desired abstention behavior.

**Audience:**

Yes

**Claims And Evidence:**

No

**Requested Changes:**

I'm not entirely familiar with the entire body of work I linked to above (https://www.connectedpapers.com/main/19800548837a32bacd4113a8d69b0e9e122be097/Calibrating-Language-Models-with-Adaptive-Temperature-Scaling/graph), but given many of those papers focus on proposing a new method rather than a thorough evaluation, I would like to believe there exists room for this paper to be of use to the community as a replication study, but in my opinion, it needs some extension of the scope to serve that purpose, and ideally, its contribution would be narrated in that fashion in writing.

RC1. Please update the narrative of the paper to reflect the existing body of work as per W1.

RC2. Please consider extending the evaluation to the corresponding non-chat/base LLMs and compare them across that axis as well (both calibration and AUROC).
One question I'm asking myself (which might have been answered already and I simply haven't yet come across the papers answering it) is how does the RLHF finetuning method (and even more interestingly/timely RLVR) impacts the loss of calibration, and whether that can be prevented? How does it impact the selective-prediction results?
Skimming the titles of some of the papers in the link suggests those questions might have already been partially answered, so adjust accordingly.

**Strengths And Weaknesses:**

Strengths:

S1. The evaluation of the chat LLMs is relatively comprehensive.

---

Weaknesses:

W1. Omission of related work and misconstruing the landscape of the field.
The paper says "The most relevant prior paper is Kadavath et al. (2022)", but the last two years saw a large number of papers evaluating the calibration in LLMs and proposing methods improving it (often via variants of temperature-scaling) - please see https://www.connectedpapers.com/main/19800548837a32bacd4113a8d69b0e9e122be097/Calibrating-Language-Models-with-Adaptive-Temperature-Scaling/graph for a non-exhaustive list of examples.
Currently, the paper does not engage with this body of work.
I think this is particularly important for TMLR, because I think the current portrayal of the research landscape borders being "misleading".

W2. Limited novelty (I appreciate this is not a criteria for TMLR acceptance).
Related to the above, the paper's findings are effectively a reproduction/corroboration of the findings of a large body of previous work.

---

> ### Author Response · Authors · 2025-06-10
> **Response to Reviewer zxok**
>
> We thank the reviewer for helpful feedback. See below for responses to each weakness and requested change.
>
> W1/RC1. We thank the reviewer for pointing out this missing related work. We will make sure to cover related work thoroughly in the revised version. We believe that the primary area of overlooked related work is post-hoc calibration methods for chat LLMs, i.e., work which verifies miscalibration of chat LLMs and studies methods for improving that calibration. Among others, two examples we will make sure to include are https://arxiv.org/abs/2409.19817 and https://arxiv.org/abs/2404.02655. We will also discuss prior work on verbalized uncertainty (as mentioned by Reviewer PGqA). Please let us know if you believe there is additional related work that needs to be included.
>
>    We will also update the narrative of the paper accordingly. In particular, we will reposition most of our results as rigorous validation of previously observed phenomena, rather than wholly new findings. The exception is our finding that correctness prediction AUROC improves as model capability improves (this is referred to as “Claim 2” in the thread in response to Reviewer PGqA.) To our knowledge, this finding had not been previously shown.
>
> W2. This point is being discussed in a thread with Reviewer PGqA, which the present reviewer (Reviewer zxok) has also commented on. To consolidate discussion of this point, we will respond to that comment thread.
>
> RC2. As discussed in the paper, evaluating the base models consistently was not possible in our prior experimental setup due to them failing to respond in the correct format. However, Reviewer PGqA recommended a new experimental setup that does potentially allow for the inclusion of the base models. There still exist some challenges associated with base models: for example, if the model assigns very low probability to all answer options, it’s unclear how trustworthy those results are. However, we will include the base models in the revised version of the paper regardless (with the appropriate caveats, if necessary).
>
> Please let us know if you have any follow-up questions or if our proposed changes do not fully address your concerns.

---

### Author Response · Authors · 2025-06-19
**Revision in progress**

We wanted to provide a quick update that we are hard at work on the revision, but since some of the requested changes require running additional experiments, the revision is taking some time. We hope to have the revision complete within 1 week (June 26), and certainly within 2 weeks (July 3).

---

### Author Response · Authors · 2025-06-25
**Revision now available**

We would like to genuinely thank all reviewers for providing such valuable and constructive feedback. We have performed a major revision of the manuscript in response to this feedback.

### **Summary of major changes**

1. Expanding the scope to include base models.
2. Changing how the MSP is computed to align with best practices. This change was also crucial for including base models since strict format-following is no longer required.
3. Expanding the related work section to provide a more thorough and accurate credit assignment to prior work.
4. Adding several new analyses, including a simple post-hoc calibration analysis, alternative uncertainty metrics (margin and entropy), and how the amount of training data affects our abstention results.

See below for more detail.

### **1. Expanded Scope and Comparison with Base Models (Reviewers zxok & PGqA)**

A key suggestion from Reviewers zxok and PGqA was to include base (i.e., non-fine-tuned) models in our analysis. We agree this is a critical comparison. We have run our full suite of experiments on 13 base models. The results are presented in the new Section 7: Experiments on base LLMs.

Consistent with Kadavath et al. (2022), base models are much better calibrated than chat models, and the calibration of base models *does* improve as capability improves. The base models also exhibit high correctness prediction AUROC, and AUROC also improves as capability improves. Even though these findings are expected, the inclusion of base models improves the comprehensiveness of our study, which we think increases the paper’s value to the community.

### **2. Methodological Improvements (Reviewer PGqA)**

Reviewer PGqA pointed out that our original method for computing the MSP was nonstandard and prevented the evaluation of base models, since the base models struggled to follow the intended format. We have adopted the methodology agreed upon in the thread with reviewer PGqA: compute the probability that the first token of the response is each of the answer tokens ("A", "B", "C", etc.) and renormalize the answer token probabilities to sum to 1. This approach is used uniformly for all models (both chat and base). The approach is defined formally in Section 4 but also mentioned briefly in Section 1, as suggested.

### **3. Expanded discussion of related work and more accurate credit assignment (Reviewers zxok & PGqA)**
We are grateful to Reviewer zxok for pointing out a significant body of recent work we had overlooked related to calibration of chat LLMs and post-hoc calibration methods in particular. We added a new subsection to Section 2 which covers this body of work and emphasizes prior findings of miscalibration in chat LLMs. We also noted that prior work found that the calibration curves of chat LLMs are roughly monotonic, which foreshadows our result that the MSP predicts correctness.

We have also added a discussion of verbalized uncertainty in LLMs, as suggested by Reviewer PGqA.

Lastly, we updated the paper's narrative to better credit prior work and frame our contributions as a comprehensive, unified study that validates some known and suspected phenomena while also presenting some novel findings.


### **4. New analyses (Reviewers PGqA, BWQa, & zxok)**
We have added several new analyses based on reviewer suggestions:

A. Post-hoc calibration (Reviewer PGqA): We ran a simple Platt scaling analysis on the chat LLMs which is covered in Appendix D. The hypothesis was that if correctness prediction AUROC improves with capability (i.e., Q&A accuracy), and we can restore calibration, then post-Platt-scaling calibration error would improve with capability as well. Unfortunately, this hypothesis did not hold. Platt scaling significantly improved calibration error, but did *not* restore the correlation with capability. This suggests that the fine-tuning causes a more complex disruption to MSPs than simply rescaling them, which could be an interesting avenue for future work.

B. Other uncertainty metrics (Reviewer BWQa): In Appendix C, we repeat the AUROC analysis for margin (difference between two largest probabilities/logits) and entropy. One interesting finding is that for logits specifically, both the AUROC values and the strength of the correlation with Q\&A accuracy are much higher for the margin compared to the maximum. This effect does not exist for probabilities. Our theory is that the probabilities are already normalized, while the logits are not, and the margin sort of functions as normalization.

C. Effect of amount of training in abstention experiments (Reviewer BWQa). We added a visualization of how the abstention score varies with the number of training samples $k$ in Appendix A.2 (Figure 10). We also added dataset-level versions of these graphs in Appendix E. These graphs validate the trend we previously mentioned where there are significant diminishing returns after k=20.

(continued below)

---

> ### Author Response · Authors · 2025-06-25
> **Revision now available (continued)**
>
> (continued from above)
>
> ### **Other changes**
>
> 1. Added Table 1 which breaks down our results in a clearer way, as suggested by Reviewer PGqA. In particular, this table more clearly distinguishes between “MSP predicts correctness” and “MSP correctness prediction improves with capability”.
> 2. Formatting fixes (Reviewer BWQa): All requested formatting changes, including moving legends outside of plots and fixing typos, have been implemented.
> 3. Added the multi-step hypothesis posed by Reviewer zxok to Section 5.
>
> We believe these revisions have substantially improved the paper's rigor and clarity. We would like to thank the reviewers again, and we would welcome any additional feedback.

---

> > ### Comment · Reviewer_zxok · 2025-06-27
> >
> > * I’ve taken a look at the revised manuscript and I am happy with the changes!
> > * Many thanks to the authors for engaging in the process, this kind of constructive process is why I like reviewing for TMLR :)
> > * One final thing (I promise) - just for the sake of curiosity:
> >
> >     1. Could you report roughly the range of parameters A & B for Platt scaling that you’ve obtained via the fitting procedure?
> >     2. Fig. 9 & Sec. 3.3 of Kadavath et al., (2022) mention T=2.5 worked well for recalibrating an RLHFed model in their case. Would you mind running your chat-LLM post-hoc calibration evaluation for a small sweep of temperature values T=[1.5, 2.5, 3.5] (i.e. remove the B parameter from Platt scaling), and comment on the results here, please? (no need to put in the manuscript unless you think it’s beneficial to the reader)

---

> > > ### Author Response · Authors · 2025-06-30
> > > **Platt scaling follow-ups**
> > >
> > > We thank the reviewer for their kind comments, and we have also appreciated their constructive engagement.
> > > 1. We’ve added the range of values for A and B to the paper. For reference, A ranges from 1.4 to 29.3 with a median of 4.1, and B ranges from -28.3 to -0.5 with a median of -3.1.
> > > 2. We ran the suggested analysis and found something interesting: although omitting B worsens the calibration (i.e., some of the gains from full Platt scaling are lost), it actually restores the correlation with Q&A accuracy. This effect held for all choices of A in {1.5, 2.5, 3.5}. We don’t think this means B should be omitted – generally one wants the best calibration possible – but we think this finding is worth adding to the Platt scaling appendix. We thank the reviewer for the suggestion.

---

> > > > ### Comment · Reviewer_zxok · 2025-06-30
> > > >
> > > > Great, many thanks!

---

> ### Comment · Reviewer_PGqA · 2025-06-30
>
> I've taken a look at the changes, and they largely satisfy my concerns. Thanks for incorporating the suggestions.

---

> ### Comment · Reviewer_BWQa · 2025-07-02
>
> I thank the authors for the revisions and additional experiments. The additional AUROC analysis for margin and additional details on the number of training samples used for calibration indeed strengthen this work.

---

### Decision · Action_Editor_nSYU · 2025-07-08

**Recommendation:** Accept as is

**Audience:**

Yes

**Audience Explanation:**

The subset of the community that cares about uncertainty quantification and LLMs would be interested in these results

**Claims And Evidence:**

Yes

**Claims Explanation:**

All reviewers agree that while the results are not super novel or surprising, the claims of the paper are backed up by evidence and are reproducible. this satisfies TMLR's criteria for acceptance.